# Terrestrial carbon isotope stratigraphy and mammal turnover during post-PETM hyperthermals in the Bighorn Basin, Wyoming, USA

Sarah J. Widlansky[1], Ross Secord[2], Kathryn E. Snell[3], Amy E. Chew[4], William C. Clyde[1]

[1]Department of Earth Sciences, University of New Hampshire, Durham, NH, 03824, USA
[2]Department of Earth and Atmospheric Sciences, University of Nebraska – Lincoln, Lincoln, NE, 68588, USA
[3]Department of Geological Sciences, University of Colorado Boulder, Boulder, CO, 80309, USA
[4]Department of Ecology and Evolutionary Biology, Brown University, Providence, RI, 02912, USA

*Correspondence to*: Sarah J. Widlansky (sarah.widlansky@unh.edu)

**Abstract.** Paleogene hyperthermals, including the Paleocene-Eocene Thermal Maximum (PETM) and several other smaller events, represent global perturbations to Earth's climate system and are characterized by warmer temperatures, changes in floral and faunal communities, and hydrologic changes. These events are identified in the geologic record globally by negative carbon isotope excursions (CIEs), resulting from the input of isotopically light carbon into Earth's atmosphere. Much about the causes and effects of hyperthermals remains uncertain, including whether all hyperthermals were caused by the same underlying processes, how biotic effects scale with the magnitude of hyperthermals, and why CIEs are larger in paleosol carbonates relative to marine records. Resolving these questions is crucial for a full understanding of the causes of hyperthermals and their application to future climate scenarios. The primary purpose of this study was to identify early Eocene hyperthermals in the Fifteenmile Creek area of the south-central Bighorn Basin, Wyoming U.S.A. This area preserves a sequence of fluvial floodplain sedimentary rocks containing paleosol carbonates and an extensive record of fossil mammals. Previous analysis of faunal assemblages in this area revealed two pulses of mammal turnover and changes in diversity interpreted to correlate with the ETM2 and H2 hyperthermals that follow the PETM. This was, however, based on long distance correlation of the fossil record in this area to chemostratigraphic records from elsewhere in the basin.

We present new carbon isotope stratigraphies using micrite $\delta^{13}C$ values from paleosol carbonate nodules preserved in and between richly fossiliferous mammal localities at Fifteenmile Creek to identify the stratigraphic positions of ETM2 and H2. Carbon isotope results show that the ETM2 and H2 hyperthermals, and possibly the subsequent I1 hyperthermal, are recorded at Fifteenmile Creek. ETM2 and H2 overlap with the two previously recognized pulses of mammal turnover. The CIEs for these hyperthermals are also somewhat smaller in magnitude than in more northerly Bighorn Basin records. We suggest that basin-wide differences in soil moisture and/or vegetation could contribute to variable CIE amplitudes in this and other terrestrial records.

# 1 Introduction

## 1.1 Paleogene hyperthermals

The early Paleogene is punctuated by numerous short duration (lasting 50–200 kyr) warming events known as "hyperthermals" (Thomas et al., 2000; Westerhold et al., 2018; Barnet et al., 2019). These hyperthermals are characterized by their rapid onsets (estimates for some range from < 10 kyr to ~20 kyr) and are associated with significant negative isotope excursions in $\delta^{13}C$ and $\delta^{18}O$ in marine sedimentary records, and negative $\delta^{13}C$ excursions in terrestrial records worldwide (e.g., Kennett and Stott, 1991; Koch et al., 1992; Zachos et al., 2001, 2008; Abels et al., 2012, 2016; Westerhold et al., 2020). Many of these hyperthermals are orbitally paced, often occurring during eccentricity maxima in 100 and 405 kyr cycles (Cramer et al., 2003; Lourens et al., 2005; Zachos et al., 2010; Dinarès-Turell et al., 2014; Abels et al., 2016; Barnet et al., 2019; although see D'Onofrio et al., 2016 for debate about which carbon isotope variations during the Paleogene constitute a hyperthermal). The largest and best known of these hyperthermals is the Paleocene–Eocene Thermal Maximum (PETM) at ~56 Ma, during which time global temperatures increased between 5–8° C (McInerney and Wing, 2011). Along with this warming came deep-ocean acidification and carbonate dissolution, and the extinction of 30–50% of benthic foraminifera (Thomas, 1998, 2007; Zachos et al., 2005; Speijer et al., 2012). Conversely, planktic foraminifera experienced geographic range shifts and proliferation over the same interval (Lu and Keller, 1995; Thomas and Shackleton, 1996; Kelly et al., 1998; Gibbs et al., 2006; Agnini et al., 2007; Mutterlose et al., 2007).

The negative carbon isotope excursions (CIEs) associated with the hyperthermals indicate massive injections of isotopically light carbon into the atmosphere, but the source and mechanism of release of these greenhouse gases is unresolved (Higgins and Schrag, 2006). Proposed sources of carbon for the PETM include destabilized methane clathrates (Dickens et al., 1995, 1997; Katz et al., 1999), volcanism associated with the North Atlantic Igneous Province (NAIP) (Gutjahr et al., 2017), thermogenic methane released from organic-rich sediments during NAIP emplacement (Svensen et al., 2004; Frieling et al., 2016), Antarctic permafrost thaw (DeConto et al., 2010, 2012), wildfires burning Paleocene peat deposits (Kurtz et al., 2003; Moore and Kurtz, 2008), and evaporation of epicontinental seas leading to the oxidation of organic matter (Higgins and Schrag, 2006), although geological evidence supporting the last two hypotheses is weak. It is also possible that multiple mechanisms and sources may have acted together and that the drivers may not have been the same for the PETM and smaller hyperthermals. Although the initial trigger for many hyperthermals may be related to orbital cyclicity, the primary cause of warming is likely the associated release of greenhouse gases. Of the smaller hyperthermals, those immediately following the PETM are currently the best studied. These include the Eocene Thermal Maximum 2 (ETM2) at ~54 Ma, also referred to as ELMO (Lourens et al., 2005) or H1 (Cramer et al., 2003), as well as the succeeding H2, I1, and I2 hyperthermals. Like the PETM, these hyperthermals are associated with negative CIEs, increased $CaCO_3$ dissolution, and foraminiferal turnover (Agnini et al., 2009; Stap et al., 2009, 2010; Jennions et al., 2015; Arreguin-Rodriguez and Alegret, 2016; D'Onofrio et al., 2016). The ETM2, H2, and I1 hyperthermals have also been linked to increased radiolarian abundance as well as shifts in calcareous plankton assemblages. Together these suggest upper water column warming, weakening thermal stratification in the upper water column, and

increased nutrients in surface waters during the events (D'Onofrio et al., 2016). The high-resolution marine record of the PETM and subsequent hyperthermals has led to a growing understanding of their effects in the ocean. A similar level of detail has not been developed for the terrestrial record of the smaller (non-PETM) hyperthermals. Achieving this level of understanding will require geochemical records from a variety of depositional environments and locations to fully understand spatial heterogeneity and underlying carbon cycle–climate dynamics. Furthermore, understanding the effects of these hyperthermals on the terrestrial ecosystem requires their recognition in richly fossiliferous strata, where both plant and vertebrate fossils occur in abundance.

## 1.2 Terrestrial record of hyperthermals

Terrestrial stratigraphic records of the PETM have been reported from North America (e.g., Koch et al., 1992, 2003; Bowen and Bowen, 2008; Baczynski et al., 2013), South America (e.g., Jaramillo et al., 2010), Europe (e.g., Cojan et al., 2000; Schmitz and Pujalte, 2007), Asia (e.g., Bowen et al., 2002; Chen et al., 2014), India (e.g., Samanta et al., 2013), and Australia (e.g., Greenwood et al., 2003). Of these, the Bighorn Basin of Wyoming preserves the most detailed, richly fossiliferous, and best studied terrestrial record of the PETM. Here the PETM corresponds with rapid turnover in both plants and mammals, as the Paleocene flora is replaced by a dry tropical flora (Wing et al., 2005) and the first representatives of several mammalian clades disperse among the Holarctic continents (Gingerich, 2006). The "dwarfing" of about 40% of mammalian lineages also occurred during the PETM (Gingerich, 1989; Clyde and Gingerich, 1998; Secord et al., 2012). The hydrologic cycle was also strongly affected (Wing et al., 2005; Foreman et al., 2012; Kraus et al., 2013; Foreman, 2014; Baczynski et al., 2017). A comprehensive summary of terrestrial environmental changes associated with the PETM can be found in McInerney and Wing (2011). In contrast to the PETM, terrestrial records of the post-PETM hyperthermals are limited. They include carbon isotope records from coal exposures in NE China (Chen et al., 2014) and western India (Clementz et al., 2011; Samanta et al., 2013; Agrawal et al., 2017), as well as records from paleosol carbonates from Europe (Cojan et al., 2000; Honegger et al., 2020), the Tornillo Basin of Texas (Bataille et al, 2016; 2019), and the McCullough Peaks region in the central Bighorn Basin (Abels et al., 2012, 2016; D'Ambrosia et al., 2017). Four post-PETM hyperthermals (ETM2, H2, I1 and I2) are preserved in the McCullough Peaks sequence and this area offers one of the best opportunities for constructing a highly detailed terrestrial record that can also be compared with marine and terrestrial records of the PETM and post-PETM hyperthermals worldwide. The McCullough Peaks fauna, however, is limited compared to other parts of the Bighorn Basin, complicating direct correlation between hyperthermals and faunal change.

Records of the PETM show that terrestrial proxies generally record a larger magnitude CIE than marine proxies, leading some to consider the terrestrial record "amplified" (Bowen et al., 2004). In marine carbonate records of the PETM, the CIE generally ranges from ~2–4 ‰ (McInerney and Wing, 2011). In terrestrial records, the PETM CIE ranges from ~3–7 ‰ depending upon the proxy used and can show considerable spatial variability depending on local environmental differences (Bowen and Bowen, 2008). Of the terrestrial $\delta^{13}C$ proxies, paleosol carbonates record the largest CIE, generally ranging from ~3–7 ‰ (e.g., Koch et al., 2003; Bowen et al., 2004; Bowen and Bowen, 2008), with a mean of ~5.5 ‰ (McInerney and Wing, 2011). A direct

comparison of marine and paleosol carbonate records indicates that the PETM CIE is on average 2.8–3.0 ‰ greater in paleosol carbonates (McInerney and Wing, 2011).

Detailed paleosol carbonate records of the post-PETM hyperthermals are limited, but also seem to show a larger magnitude
CIE relative to the marine record. Marine benthic carbonate records of the ETM2 and H2 CIEs are ~1.4–1.5 ‰ and 0.8 ‰, respectively (Stap et al., 2010; Barnet et al., 2019), whereas the terrestrial ETM2 and H2 records from soil carbonates in McCullough Peaks are ~3.8 ‰ and ~2.8 ‰ (Abels et al., 2012, 2016; D'Ambrosia et al., 2017). No terrestrial organic carbon records exist for these hyperthermals from the Bighorn Basin, although bulk organic carbon records from India and China have an average magnitude of ~2.7–3.5 ‰ for ETM2 and ~2.5 ‰ for H2 (Chen et al., 2014; Agrawal et al., 2017). Moreover, there
is debate surrounding the scaling relationship between the amplitude of the PETM and post-PETM CIEs, with some arguing for a consistent linear relationship and therefore common source (e.g., Chen et al., 2014) and others arguing that the post-PETM CIEs appear to scale linearly with each other, but not with the PETM, potentially suggesting a different triggering mechanism for the later hyperthermals (Abels et al., 2016).

The cause of the apparent "amplification" in terrestrial CIEs is not fully understood, although increased soil productivity
(Bowen, 2013), increased humidity (Bowen et al., 2004; Bowen and Bowen, 2008), increased carbon isotope discrimination by plants under higher atmospheric $p$CO2 conditions (Schubert and Jahren, 2013), and the loss of gymnosperms in the PETM (Smith et al., 2007), which have higher $\delta^{13}C$ values than angiosperms, have been invoked as potential mechanisms. Alternatively, increased dissolution during hyperthermals could produce a muted CIE signal in some marine records relative to actual atmospheric $\delta^{13}C$ shifts (Zachos et al., 2005). The limited number of terrestrial records of post-PETM hyperthermals
outside of the McCullough Peaks area makes it difficult to understand how these records vary spatially and to compare CIE magnitudes between the terrestrial and marine realms.

### 1.3 Mammal response to hyperthermals

The PETM coincides with one of the most profound phases of mammal turnover during the Cenozoic, marked by the first appearances of several clades including: Perissodactyla, Artiodactyla, Euprimates, and Hyaenodontidae (e.g., Gingerich, 1989,
2003, 2006; Koch et al., 1992; Clyde and Gingerich, 1998; Hooker, 1998; Rose et al., 2012). These taxa disperse among North America, Europe, and Asia during the PETM, but their areas of origin are uncertain (e.g., Bowen et al., 2002; Clyde et al., 2003; Gingerich, 2006; Smith et al., 2006; Morse et al., 2019). Several groups also demonstrate transient "dwarfing" during the PETM (Gingerich, 1989; 2006; Clyde and Gingerich, 1998; Strait, 2001; Chester et al., 2010), with minimum body sizes corresponding to peak warming in Bighorn Basin equids (Secord et al., 2012).
Previous work on mammal faunas from post-PETM hyperthermals in McCullough Peaks suggests that mammal body size decreased during ETM2 in at least two lineages, though the fauna did not show major reorganization as is seen during the PETM (Abels et al., 2012; D'Ambrosia et al., 2017). However, PETM reorganization may largely be related to intercontinental and intracontinental immigrants making first appearances, while there is no strong evidence for such immigrants appearing

around the time of ETM2 or H2 (Woodburne et al., 2009; Chew, 2015). Until now, the lack of abundant fossil mammals tied closely to the stable isotope record has precluded a detailed assessment of mammal faunal change through these hyperthermals. Chew (2015) identified two distinct pulses of increased turnover and diversity, known as faunal events B–1 and B–2, in mammal assemblages from the Fifteenmile Creek area in the south-central Bighorn Basin. In contrast to the PETM, faunal change during these events was lower in magnitude and driven by increased beta, rather than alpha, diversity. Moreover, the number of first appearances was roughly equal to the number of last appearances and no known intercontinental immigrant taxa arrived during the B–1 and B–2 events. Additionally, abundance shifts during these events appear to have favored smaller bodied species (Chew, 2015). Species richness parameters used by Chew (2015) were significantly correlated to sample sizes, potentially indicating some sample size bias. Chew and Oheim (2013), however, found similar increases in beta richness after correcting for sample variation in the same study area, suggesting that these patterns are independent. Chew (2015) suggested that the ETM2 and H2 hyperthermals were potential drivers of faunal change during events B–1 and B–2, based on inferred temporal overlap of these events with geochemical records from other parts of the basin. Preliminary geochemical work in this area, however, did not identify any clear CIEs in this stratigraphic interval that would confirm this connection (Koch et al., 2003).

Our work presents new, high-resolution carbon isotope records from paleosol carbonates through the Fifteenmile Creek area in the south-central Bighorn Basin. This allows, for the first time, the extensive fossil mammal records from this area to be tied directly into the isotope stratigraphy of post-PETM early Eocene hyperthermals. The results offer a direct test of the hypothesis that faunal turnover events B–1 and B–2 are correlated with the ETM2 and H2 hyperthermals (Chew, 2015). This new record also adds to the terrestrial stable isotope record of post-PETM hyperthermals in the Bighorn Basin, allowing for better spatial characterization of these events.

## 2   Geological setting

The Bighorn Basin is a Laramide structural basin located in northwest Wyoming. Uplift of the Beartooth, Pryor, Bighorn, and Owl Creek mountains resulted in sediment accumulation in the basin during the early Paleogene (Gingerich, 1983; Kraus, 1992). The early Paleogene basin axis was located near the western margin of the basin and sediment thickness generally decreases going from the northwest to southeast within the basin (Parker and Jones, 1986). Regional differences in subsidence are potentially due to activation of east–west trending buried faults (Kraus, 1992). The lower Eocene Willwood Formation represents a fluvial depositional system, and it is typified by paleosols of varying maturity, channel sandstones, crevasse-splay deposits, and localized carbonaceous shales (Kraus, 1992, 1997; Bown and Kraus, 1993). The most complete published record of early Eocene hyperthermals comes from the McCullough Peaks area. This area experienced rapid subsidence during this time and the Willwood Formation here is characterized by a thick sequence of relatively immature paleosols.

The Fifteenmile Creek area of the Bighorn Basin is located ~80 km southeast of McCullough Peaks and, in contrast to the McCullough Peaks, it is characterized by having more mature paleosols, as well as more channel and cut-and-fill deposits

associated with slower subsidence and aggradation. The region has produced an extensive collection of nearly 1000 mammal fossil localities that have been tied to a 700 m thick composite stratigraphic section (Bown et al., 1994). Previous work in the area suggested that the stratigraphic interval containing the ETM2 and H2 hyperthermals was between 380 and 455 meters in the Bown et al. (1994) composite section (Chew, 2015). This interval occurs in the *Bunophorus etsagicus* mammalian interval

zone (Wa-5), following the last occurrence of *Haplomylus speirianus*. The *Bunophorus etsagicus* interval zone contains the faunal event Biohorizon B (Schankler, 1980; Bown et al., 1994; Chew, 2009), which occurs just below ETM2 and H2 in sections farther north (Abels et al., 2012; D'Ambrosia et al., 2017). The Chron C24r – C24n.3n geomagnetic polarity reversal provides additional support for this interval containing ETM2 and H2. In McCullough Peaks and other records around the world, the C24r – C24n.3n reversal occurs just above ETM2 (Lourens et al., 2005; Abels et al., 2012; D'Ambrosia et al., 2017).

Thus, ETM2 is closely bracketed by Biohorizon B below and the Chron C24r – C24n.3n polarity reversal above. Magnetostratigraphic work from Elk Creek Rim, about 20 km north of the Fifteenmile Creek sections studied here, placed the C24r – C24n.3n reversal at the ~450 meter level of the Bown et al. (1994) composite section, ~70 meters above Biohorizon B (Clyde et al., 2007; however see Tauxe et al., 1994 for earlier magnetostratigraphic interpretation). Based on these constraints, we constructed detailed carbon isotope datasets from stratigraphic sections along Fifteenmile Creek that span the critical

interval between Biohorizon B and the C24r – C24n.3n reversal to identify ETM2 and H2.

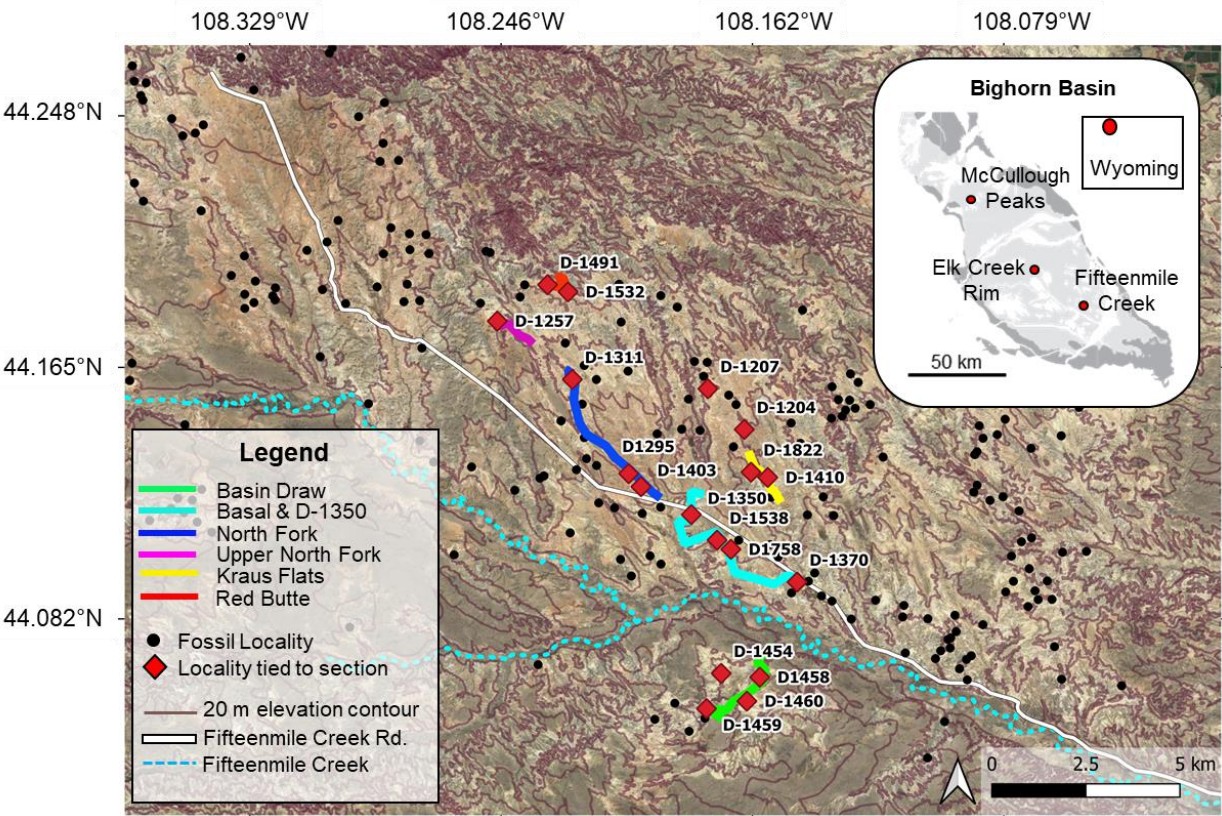

**Figure 1: Map of the Fifteenmile Creek area of the Bighorn Basin, WY showing mammal fossil localities (black dots) and carbon isotope sections (colored lines), overlain onto satellite imagery. Fossil localities that were sampled directly or tied to these sections via lithological tracing are shown as red diamonds and labelled with locality numbers. Inset map shows the location of the Fifteenmile Creek area relative to McCullough Peaks and Elk Creek Rim within the Bighorn Basin. Shading key for the inset map: dark grey = Fort Union Formation (Paleocene), light grey = Willwood Formation (Eocene), white = Quaternary alluvium. Basemap source: Google Earth © 2021. More detailed maps of the individual sections are available in Figures A1 – A6.**

## 3    Methods

### 3.1 Field methods

Pedogenic carbonate nodules were collected from six primary stratigraphic sections that span the target interval (Fig. 1). All samples were collected from fresh, unweathered rock and at least three nodules were analyzed from each sampling site, when possible. Sample sites were chosen to ensure at least a 1-meter sampling resolution, where possible, and levels near a potential CIE were sampled at a higher resolution during subsequent field seasons. Samples were collected from > 30 cm below the upper surface of B horizons in variegated paleosols. At this depth, soil $CO_2$ is primarily plant-respired and will track atmospheric $CO_2$ $\delta^{13}C$ (Cerling, 1984). The relative stratigraphic positions of sampling sites and fossil localities were measured in local sections using a Jacob's staff and bed tracing. Because this area it typified by low topographic relief, sections were tied together with marker bed traces, sometimes over fairly long distances. Marker beds were usually prominent, red paleosols

or the contacts between two contrasting paleosols. Sometimes a marker bed would become covered as it was traced up section and we would measure up to the next prominent marker bed and trace it into the next section. In many cases, we were able to measure back down to the original marker bed when more complete exposure was available. In some cases, the original marker bed would change color over the trace distance, and/or the surrounding beds would change color.

Our local stratigraphic sections were correlated to the composite stratigraphic meter levels of Bown et al. (1994) (hereafter referred to as "Bown composite meter levels" or BCM) using the BCM levels of fossil localities that were measured into our sections. We then used our Jacob's staff measurements to determine BCM levels for the rest of the section, based on these fossil locality tie-points (Fig. A8). Because of the difficulty of tracing individual beds over long distances in this low relief area, we used local elevation to test our lithostratigraphic correlations. Elevation is a reasonable proxy for stratigraphic level in the Fifteenmile Creek area since the dip is close to 0°. Elevations were collected using a Trimble GeoXT 6000 handheld differential GPS (dGPS) paired with a Trimble Tornado external dual-frequency antenna. Elevations were collected from sample sites and from important stratigraphic markers (e.g., marker beds and fossil producing horizons) in every section (Appendix A). Checks on local dips were made by shooting a horizontal line between exposures of a marker bed along the section transect. If local differences in dip along the transect were detected, adjustments to dip were made on the Jacob Staff's clinometer. Postprocessing of dGPS data was done using the Trimble Pathfinder Office and Terra Sync Software. After processing, the elevations typically had an accuracy of < 50 cm.

A total of 18 fossil localities, which include more than 4500 identified fossil mammal specimens, were directly tied to the six primary local stratigraphic sections from which isotope samples were collected (Fig. 1) and used to correlate these sections to BCM levels. Three shorter sections were also sampled for carbonate nodules, one near the main Basin Draw section (through fossil locality D-1454), and two located near the Kraus Flats and Red Butte sections (through localities D-1207 and D-1532, respectively) (Appendix B).

**3.2 Laboratory methods**

Individual nodules were polished using a diamond lap wheel and were visually inspected to identify any primary, secondary, or altered textures. Micritic calcite and sparite were sampled for stable isotope analysis. Micrite is generally interpreted to represent a primary phase formed by precipitation directly from soil $CO_2$, and thus is appropriate for paleoclimate reconstruction (Bowen et al., 2001). Sparite samples were also drilled from a subset of nodules to compare stable isotope values of primary and secondary components (e.g., Bowen et al., 2001; Snell et al., 2013). Sparite in paleosol carbonate nodules generally forms in veins and septarian-style cracks in the nodules. It is thought to precipitate from fluids after deep burial at high temperatures, resulting in lower $\delta^{18}O$ values than the original soil water $\delta^{18}O$ (Bowen et al., 2001; Snell et al., 2013). All stable isotope ratios are reported in delta ($\delta$) notation as per mil (‰) relative to the Vienna Pee Dee Belemnite (VPDB) standard, where $\delta = ([(R_{Sample} / R_{Standard}) - 1] \times 1000)$, and R is the ratio of the heavier mass isotope to the lighter mass isotope. Analyses of $\delta^{13}C$ and $\delta^{18}O$ values were done at the University of Michigan Stable Isotope Laboratory (UMSIL) using a Thermo Finnegan MAT253 stable isotope ratio mass spectrometer attached to a Kiel IV automated preparation device, and at the

University of Colorado Boulder Earth Systems Stable Isotope Laboratory (CUBES–SIL) using a Thermo Delta V continuous flow stable isotope ratio mass spectrometer attached to a GasBench II gas preparation device. Repeated measurements of in-house standards yield precision of ± 0.1 ‰ or better for both $\delta^{13}C$ and $\delta^{18}O$ in both laboratories, although overall uncertainties are likely slightly larger than this in carbon isotopes due to extrapolation of the standard correction lines beyond the lowest standard values for both labs (Appendix C). Data correction and calculations from CUBES–SIL were done using in-house R scripts that utilized Tidyverse and IsoVerse R packages (R Core Team, 2019; Wickham et al., 2019; Kopf et al., 2021; Widlansky et al., 2022). Powder from a subset of 10 micrite samples was homogenized and replicates were sent to UMSIL and CUBES–SIL to identify if there were differences between the two labs. $\delta^{13}C$ and $\delta^{18}O$ values measured at UMSIL were then corrected to the CUBES–SIL values based on these replicate samples (Appendix C).

CIE magnitudes were calculated by detrending the data to account for long-term early Paleogene trends, then taking the difference between the detrended $\delta^{13}C$ background values and the values from the body of the CIEs, following the method of Abels et al. (2016). $\delta^{13}C$ values were averaged from each stratigraphic level prior to calculating the magnitude, and the mean peak excursion values from each section were also averaged to facilitate comparisons to other records and account for local spatial heterogeneity.

## 4    Results

### 4.1  Stable isotopes

A total of 789 carbonate samples from the Fifteenmile Creek area were analyzed for $\delta^{13}C$ and $\delta^{18}O$ (Figs. 2 and 3). $\delta^{13}C$ values for micritic carbonate samples (n = 770) range from -16.4 ‰ to -8.7 ‰ (mean = -11.0 ‰). Micrite $\delta^{18}O$ values range from -16.6 ‰ to -6.5 ‰ (mean = -9.3 ‰). Sparite samples (n = 19) display $\delta^{13}C$ values between -21.2 ‰ and -9.5 ‰ (mean = -12.6 ‰) and $\delta^{18}O$ values between -23.4 ‰ and -8.6 ‰ (mean = -14.8 ‰) (Fig. 2). Twelve samples were removed due to relatively low weight percent carbonate (< 50 %). Low weight percent carbonate introduces the possibility that the carbonate in these samples does not reflect pedogenesis, or they could be higher porosity samples that were more susceptible to alteration. Fifty percent is a somewhat conservative cutoff that retains only samples that are majority carbonate. Considering only sampling sites with at least three nodules, the mean within–site range in $\delta^{13}C$ is 0.7 ‰ but can be up to 3.2 ‰. The mean within–site range in $\delta^{18}O$ is 0.9 ‰ and up to 5.8 ‰. Together, the within–site range in both $\delta^{13}C$ and $\delta^{18}O$ are generally < 1 ‰.

It is possible that small amounts of unrecognized spar or areas of recrystallization were drilled with the micrite, which could contribute to the large spread in $\delta^{18}O$ towards more negative values (Fig. 2). As such, samples with $\delta^{18}O$ values greater than 2σ below the mean micrite $\delta^{18}O$ value (i.e., < -11.7 ‰) were identified as potentially incorporating a secondary spar phase and are shown as such in Figs. 3 and 4. Secondary recrystallization in a closed system does not affect carbon isotopes to the same extent as oxygen isotopes (Cerling, 1984), suggesting that partially recrystallized samples may still record CIEs reliably. Two stratigraphic levels from the base of the Basin Draw section have anomalously low $\delta^{13}C$ values (up to -16.4 ‰) that fall outside

of the typical range for early Eocene pedogenic carbonate from this region and could incorporate some mixing with sparite, particularly given their relatively low $\delta^{18}O$ values (down to -12.1 ‰).

Replicate samples that were analyzed by the two laboratories (n = 10) show that $\delta^{13}C$ values differed by up to 0.7 ‰ (mean = 0.4 ‰, standard deviation = 0.3 ‰) while $\delta^{18}O$ values differed by up to 0.6 ‰ (mean = 0.3 ‰, standard deviation = 0.2 ‰). Greater differences between the two labs are consistently associated with more negative values, and $\delta^{13}C$ and $\delta^{18}O$ values produced in the CUBES–SIL lab are generally more negative than values from the UMSIL. These offsets are likely due to differences in standardization between the labs and this relationship was used to correct the UMSIL data to the CUBES–SIL

values (Appendix C). This laboratory offset is within the range of observed within–site variation described above, indicating that the offset is not likely to affect the stratigraphic interpretations relevant to this study.

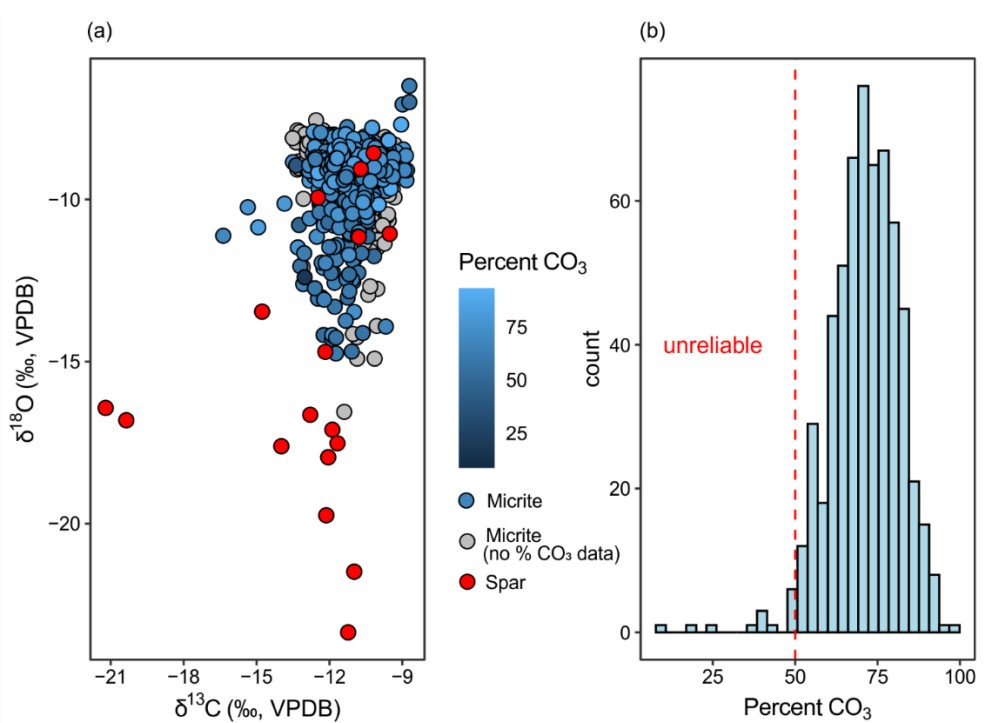

**Figure 2: (A) $\delta^{13}C$ and $\delta^{18}O$ for sparite and micrite samples measured from Fifteenmile Creek. Samples analyzed in the CUBES–**
**SIL lab are colored according to their weight % $CO_3$. Samples analyzed by the UMSIL lab are shown in grey. (B) Weight percent carbonate from the CUBES–SIL samples. Twelve samples with less than 50% $CO_3$ were considered unreliable and were removed.**

### 4.2  Carbon isotope excursions

Carbon isotope excursions were identified as places where site mean $\delta^{13}C$ values were less than -12 ‰ and the values show a clear decrease and return to background levels (between about -9 ‰ and -11 ‰, mean background $\delta^{13}C$ = -10.3 ‰). Places

where single samples or site means were less than -12 ‰ and were surrounded by background values above and below, or places where the return to background values was not captured were not considered CIEs. CIEs were identified in all six

sections. In Basin Draw, there is a CIE between the ~45 to 55 m level with minimum $\delta^{13}$C site values reaching -12.0 ‰. The lower ~35 m of this section are characterized by noisy $\delta^{13}$C records that are more negative than the average background value, but do not show a clear CIE pattern. In the Basal/D-1350 section, there is a well-defined CIE between 30 – 45 m with a

minimum mean $\delta^{13}$C site value of -13.3 ‰. Two CIEs are recorded in the Kraus Flats section, one between ~ 0 – 25 m (minimum site $\delta^{13}$C = -12.8 ‰) and another between ~35 – 45 m (minimum site $\delta^{13}$C = -12.7 ‰). The base of the North Fork section includes sites with relatively low $\delta^{13}$C values (minimum site $\delta^{13}$C = -13.3 ‰) that return to background levels around the 20 m level. Based on field correlations and close geographic proximity with the Basal/D-1350 section (Appendix A), we interpret this negative shift to be equivalent to the excursion at the top of the Basal/D-1350 section. This interpretation is also

supported by composite meter levels from Bown et al. (1994), which indicate that stratigraphic levels at the top of the Basal/D-1350 section should be equivalent to levels in the bottom of the North Fork section. Thus, we consider the negative shift in values at the base of the North Fork section to represent a CIE, even though the onset is not captured. The ~20 – 30 m level in North Fork also records a smaller CIE (minimum site $\delta^{13}$C = -12.2 ‰), though the record is less clear. The Upper North Fork section captures the onset and recovery of one CIE, with a minimum site $\delta^{13}$C of -12.2 ‰ at the 12 m level. The Red Butte

section records a CIE between 0 – 20 m (minimum site $\delta^{13}$C = -12.2 ‰) and also appears to capture the onset of a second CIE in the upper 10 m of the section. Because the recovery and potentially even the minimum values of this uppermost excursion are not present in our record, we have not identified it as a CIE, although future work may determine it is a CIE.

In addition to the correlation between the Basal/D-1350 CIE and the lowest CIE in North Fork, measured sections and bed traces indicate that the Upper North Fork CIE is stratigraphically higher than the second CIE in the North Fork section. Bed

traces between the Upper North and Red Butte sections also indicate that the CIEs in the Upper North Fork and Red Butte sections are equivalent (Appendix A). Both the Upper North Fork and Red Butte CIEs occur in lateral equivalents of the same yellow siltstone. The Kraus Flats section was not tied directly to another section by bed tracing. However, the carbon isotope stratigraphy and general placement of this section within the composite framework of Bown et al. (1994) support a correlation between the two CIEs in Kraus Flats and the two CIEs in North Fork. The Basin Draw section was also not correlated to

another section by tracing beds. The BCM levels of fossil localities in the upper part of this section support correlation to the higher CIE in North Fork and Kraus Flats, however, no lower CIE is clearly preserved in this section. Additionally, differential GPS elevations suggest that this CIE may line up better with the lower CIE in the North Fork and Kraus Flats sections (Fig. 4). For these reasons, the Basin Draw CIE remains uncorrelated. Together these data indicate that three distinct CIEs are fully recorded in the field area. The lowest CIE ("CIE1") occurs in the Basal/D-1350, North Fork, and Kraus Flats sections. The

middle CIE ("CIE2") occurs in the North Fork and Kraus Flats sections and the highest CIE ("CIE3") occurs in the Upper North Fork and Red Butte sections.

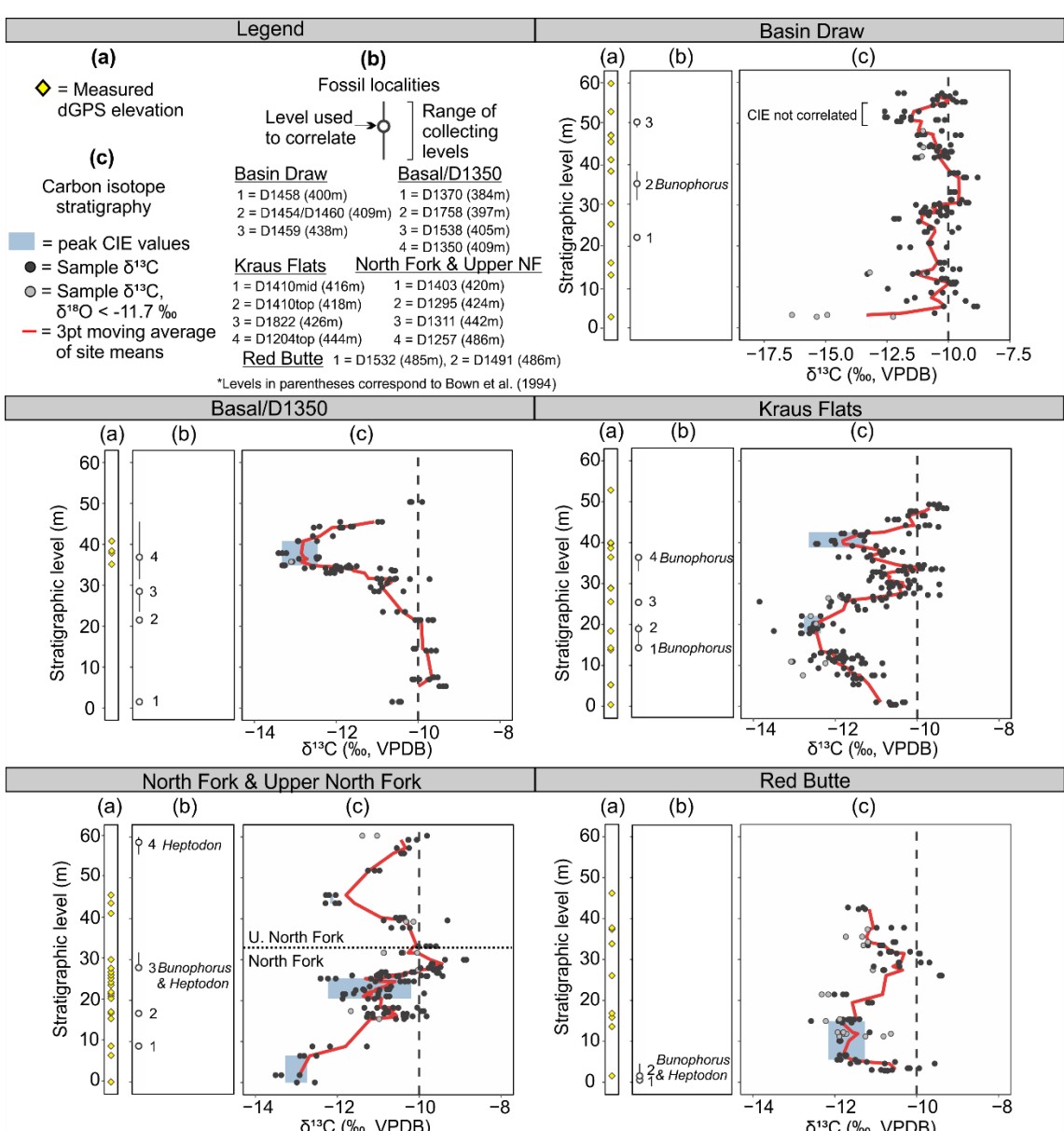

**Figure 3: Carbon isotope data from pedogenic carbonate nodules for Fifteenmile Creek sections. (a) Stratigraphic levels where dGPS elevations were collected. (b) Fossil localities that were sampled within the section. Open points correspond to the Bown et al. (1994) composite meter level for the locality and main collecting level. Lines show ranges of collecting levels. Fossil localities with *Bunophorus* are Wa5 or above. Localities with *Heptodon* are Wa6 or above. (c) Carbon isotope stratigraphy. Points represent measurements for individual nodules (black = $\delta^{18}O > -11.7$ ‰, grey = $\delta^{18}O < -11.7$ ‰). Red lines are a three-point moving average through the mean values from a sampling site and blue shading shows range of peak values used to calculate CIE magnitudes. Note that the CIE at ~45 m in the North Fork/Upper North Fork sections includes a small range of ~-12 ‰ values, making the blue shading there difficult to see. Black dashed lines at -10 ‰ show typical background value for Fifteenmile Creek.**

## 4.3 Stratigraphy

Placing the individual measured sections into the BCM framework can help confirm the field-based correlations (Fig. 4). CIE1 occurs between the 405 – 430 BCM levels, CIE2 between the 430–455 BCM levels, and the CIE3 between the 460 – 500 BCM levels (Fig. 4a). The dGPS elevations offer an additional check on these correlations. When the data are plotted in a dGPS elevation framework, CIE1 occurs between 1320 – 1340 m in elevation, CIE2 between 1340 – 1355 m, and CIE3 between 1370 – 1405 m (Fig. 4b). Oxygen isotopes do not show any clear differences between hyperthermal and non-hyperthermal intervals (Appendix D). This is consistent with observations that changes in pedogenic carbonate $\delta^{18}$O during the PETM are muted compared to the $\delta^{13}$C record, possibly due to temperature fractionation in the carbonate acting in an opposing direction to temperature fractionation in the atmosphere (e.g., Koch et al., 2003). The magnitudes of the CIEs also vary, both across the different sections and from one CIE to another. The average magnitude of CIE1 is 3.1 ‰. The average magnitude of CIE2 is 1.7 ‰ and the average magnitude of CIE3 is 2.1 ‰.

The measured thicknesses of the local sections are different than the thicknesses from BCM levels and those inferred from elevation (Fig. 4). The Basin Draw section (55 m thick) is shortened to 41 m in the elevation framework and stretched to 64 m in the BCM framework. The Basal/D-1350 section (49 m thick) maintains a similar thickness in the elevation framework and is shortened to 38 m in the BCM framework. The Kraus Flats section (49 m thick) is shortened to 31 m in the elevation framework and stretched to 55 m in the BCM framework. The Red Butte section (40 m thick) maintains similar thicknesses in both the BCM and elevation frameworks. The North Fork and Upper North Fork sections were correlated to each other using a bed trace in the field (Figs. 3, A4, and A5). The thickness measured for both sections together, based on this bed trace, is ~60 m (Fig. 3), 76 m separate the base of the North Fork section and the top of the Upper North Fork section in the elevation framework and there is 75 m separating them in the BCM framework (Fig. 4). See Appendix A for elevation and fossil locality tie points used in each section.

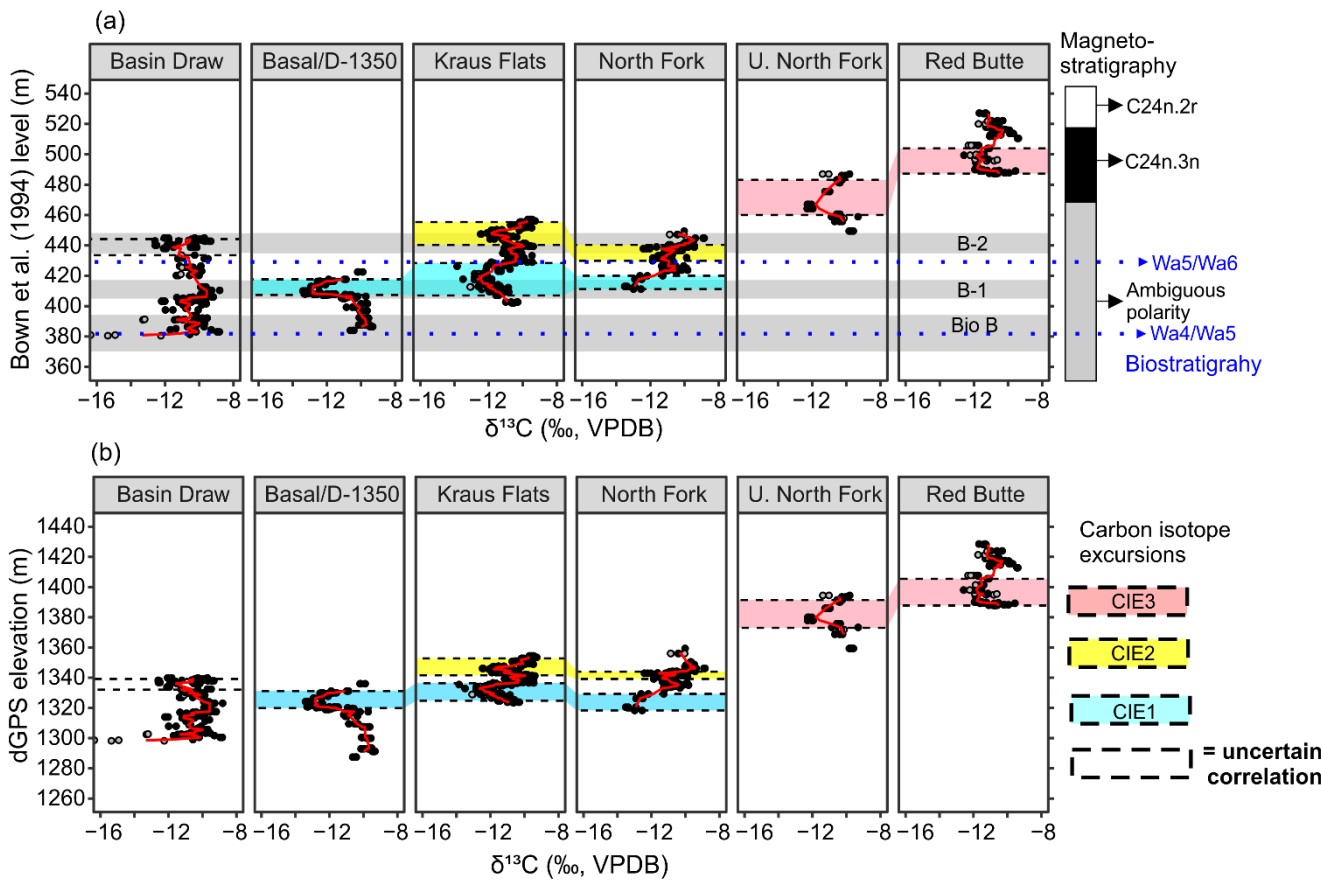

**Figure 4: Carbon isotope data for Fifteenmile Creek sections shown using the (a) Bown et al. (1994) equivalent meter level, and (b) differential GPS elevation. The stratigraphically lowest, middle and highest carbon isotope excursions are outlined by the blue, yellow and red shading, respectively. The stratigraphic intervals associated with the faunal events described in Chew (2015) are shown as grey shaded regions in the top panel, along with faunal zone boundaries (blue dotted lines) based on Bown et al. (1994). Magnetostratigraphy is based on Clyde et al. (2007), partly reinterpreted from Tauxe et al. (1994). Points represent measurements for individual nodules (black = δ18O > -11.7 ‰, grey = δ18O < -11.7 ‰). Red lines are a three-point moving average through the mean values from a sampling site.**

## 4.4 Fossil Mammal Localities

Chew (2015) proposed that the 405 – 417 and 435 – 448 BCM levels could correspond to the ETM2 and H2 hyperthermals, based on the observed changes in mammal diversity and rate of turnover in these intervals. The new Fifteenmile Creek isotope records presented here generally support this hypothesis, with CIE1 overlapping with the 405 – 417 m interval (faunal event B–1) and CIE2 overlapping with the 435 – 448 m interval (faunal event B–2) (Figs. 4 and 5). This is especially apparent in the Basal/D-1350, Kraus Flats, and North Fork sections. The D-1350 mammal locality (408 – 410 BCM level), for example, shows a clear CIE, and also yields several species that first appear during faunal event B–1 (e.g., *Xenicohippus grangeri*, *Diacodexis secans*, *Didymictis lysitensis*, *Eohippus angustidens*). Localities D-1410 (410 – 418 BCM level) and D-1204 (438 – 444 BCM level), which were tied into the Kraus Flats section, produce faunas that are consistent with the B–1 (D-1410) and B–2 (D-

1204) faunal events of Chew (2015), and also overlap with the CIE1 and CIE2 intervals, respectively. However, some localities show small offsets from their predicted position relative to the CIE record. Locality D-1311 (442 BCM level), for example, falls within the BCM interval associated with faunal event B–2, but is just above CIE2. Locality D-1403 (420 BCM level) is within CIE1 but falls outside of the BCM interval reported to contain faunal event B–1 (Fig. 5).


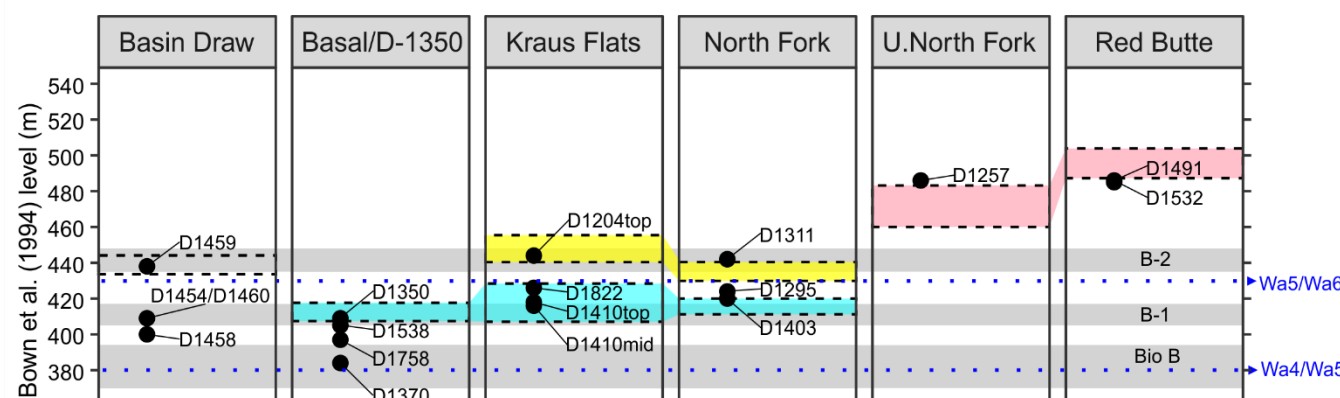

**Figure 5: Selected Fifteenmile Creek fossil localities tied to the Bown et al. (1994) composite section. Shading follows Fig. 4, where stratigraphically lowest, middle, and highest carbon isotope excursions are outlined by blue, yellow and red shading, respectively. The stratigraphic intervals associated with the faunal events described in Chew (2015) are shown as grey shaded regions. Points for the fossil localities incorporate some stratigraphic uncertainty and averaging of collecting levels discussed in the text. Faunal zone boundaries are shown as blue dotted lines based on Bown et al. (1994). For more detailed ranges of collecting levels associated with the localities, see Fig. 3.**


## 5    Discussion

### 5.1  Stratigraphic correlation and uncertainty

Several aspects of the Fifteenmile Creek stratigraphy complicate direct correlation between sections in the study area, including large lateral spacing and relative lack of vertical relief of exposures. To account for this, we used two methods of correlation, traditional bed tracing and correlation to the BCM framework, and elevations based on dGPS to supplement our field observations. Each of these methods of correlation has its own inherent uncertainty. For example, early fossil collecting in the region may not have specified the producing level for each fossil specimen. The BCM levels therefore reflect some degree of

stratigraphic averaging of fossil horizons. These composite levels also rely on bed tracing, which is difficult to do precisely over long distances in this low-relief terrain. Although the dGPS elevations presented here account for some of this uncertainty in bed tracing, they assume perfectly horizontal bedding and estimates of thicknesses can be erroneous if there are changes in local dip or any structural displacement. Although dip is generally close to zero in this area, bed tracing combined with dGPS

measurements has highlighted some local variation in bedding attitude in the field area. The elevations also may be affected by variations in paleo-topography, including lateral variation in paleosol thickness due to relative position on the floodplain and channel meander. Paleosol and sandstone thickness can vary up to a meter in this area, which is larger than the vertical error of the dGPS (~50 cm). Vertical dGPS error on the elevations therefore likely contributes very little to the observed variation among sections. Field correlation between sections has a higher uncertainty of up to 10 m and in some cases was not possible (Appendix A). Correlation between subsections within a section has a much lower uncertainty and is generally < 1 m for shorter bed traces (Appendix A).

Many aspects that complicate stratigraphic correlation in this region are illustrated with the Basin Draw section. This section is located south of Fifteenmile Creek, across from the main portion of the composite section (Fig. 1) making direct lithostratigraphic correlations difficult. Exposures along this section are also spaced far apart and dGPS elevations routinely underestimate thickness compared to bed tracing between exposures here, suggesting that dip and/or paleosol thickness variation may disproportionally affect these longer traces. Further complicating correlation, a large cut-and-fill deposit occurs in the upper portion of the Basin Draw section and fossil locality D-1459 (between ~430–440 BCM level, or 1330–1336 m elevation) is in the cut-and-fill (Fig. A1).

Despite uncertainties, stratigraphic levels for each of the three CIE intervals are in better agreement across the six sections using dGPS elevation, than with BCM levels (Fig. 4). Using elevation, the CIE in the Basin Draw section correlates better with the stratigraphically lowest excursion in other sections, particularly when we exclude the outliers at the base of the section that appear to be diagenetically altered and not associated with a well-defined excursion. The local biostratigraphy rules out the possibility that these basal samples could be capturing the PETM and the very negative carbon and oxygen isotope values are more consistent with diagenesis. The Basin Draw section also does not preserve two clear CIEs, suggesting that the CIE near the top of the section would correlate to CIE1 in our record. It should be noted that the faunal analysis of Chew (2015) places the top of the Basin Draw section within faunal event B–2, and therefore predicts that it would overlap with CIE2 which is inconsistent with this correlation. Due to the ambiguous correlation between Basin Draw and the other sections, we do not correlate this CIE to either of these intervals and its correlation remains unresolved.

## 5.2 Identification of post-PETM hyperthermals

To account for the geographic variability in bed thickness described above, the five stratigraphic sections that are north of Fifteenmile Creek (Basal/D-1350, Kraus Flats, North Fork, Upper North Fork, and Red Butte), as well as the shorter sections through fossil localities D-1207 and D-1532, were compiled into a "Fifteenmile Creek Composite Section" (Fig. 6). To create the composite section, the peak negative $\delta^{13}C$ values from each CIE were aligned and the relative spacing between CIEs was scaled to the average of the spacing from the sections where the CIEs were measured together. All stratigraphic thicknesses for the composite section are based on the thicknesses measured using a Jacob's staff in the field, rather than using BCM levels or elevations. The Jacob's staff measurements offer the least uncertainty for individual sections and isotope values are the best means for correlating between sections for purposes of creating a composite section. These correlations are independently

supported by both the BCM and elevation frameworks. Using this composite section, CIE1 occurs between the ~35 to 55 meter level. CIE2 occurs between the ~57 to 70 meter level. CIE3 occurs between the ~85 to 104 meter level, with some relatively low δ¹³C values above this that could possibly represent the onset of a fourth CIE (Fig. 6). We confidently identify CIE1 and CIE2 as the ETM2 and H2 hyperthermals, respectively, based on their position relative to Biohorizon B. Elsewhere in the Bighorn Basin, ETM2 lies just above Biohorizon B (Abels et al., 2012; D'Ambrosia et al., 2017). Biohorizon B occurs at the ~11 meter level in our new composite section, suggesting the lowest CIE that starts at ~35 meters is ETM2 (Fig. 6). Magnetostratigraphic data are consistent with this correlation, as the Chron C24r – C24n.3n polarity reversal is known to lie just above ETM2 and correlates to somewhere in the lower 100 meters of our new composite section (Tauxe et al., 1994; Clyde et al., 2007) (Fig. 6).

Identification of the stratigraphically highest CIEs is more difficult. One possibility is that they represent I1 and the onset of I2. Another possibility is that they correspond to a less well-defined negative CIE between H2 and I1 near the 260 m level in the McCullough Peaks composite section that was attributed to local environmental factors (Abels et al., 2016) (Fig. 6). Barnet et al. (2019) recently showed a small CIE at roughly the same level in the marine record, suggesting this could be a global signal. Biostratigraphy does little to help resolve these options, due to the small fossil sample sizes in McCullough Peaks. The magnetostratigraphic record supports correlation of our highest CIEs to I1 and I2, based on the presence of the C24n.3n – C24n.2r reversal near the top of our section which is often recorded just after I2 in other records (e.g., Zachos et al., 2010). It is important to note that our sample sections were not correlated directly to the Tauxe et al. (1994) magnetostratigraphy for Fifteenmile Creek. Instead, the magnetostratigraphy is linked with our data using the BCM levels for both records. Moreover, the magnetostratigraphy we've presented in Fig. 6 is a combination of Clyde et al. (2007), who offer a revised level for the C24r – C24n.3n reversal from Elk Creek Rim ~25 km north, and Tauxe et al. (1994), who constrain the level for the C24n.3n – C24n.2r reversal from Fifteenmile Creek. The stratigraphic spacing between the lower and higher CIEs in our section supports the second scenario (correlation to the poorly defined negative CIE between H2 and I1) supporting the idea that this isotopic signal may be more regional or global in scale (Fig. 6). It is also possible that there is a hiatus in the upper part of the Fifteenmile Creek section or that sediment accumulation rates were much slower in this interval. In either case, CIE3 could represent I1.

Cyclostratigraphic constraints on depositional rates across the basin support correlation of CIE3 to the interval between H2 and I1. Aziz et al. (2008) reported a depositional rate of ~2.5 kyr/m at their Red Butte section, located ~3 km northwest of our Red Butte Section, which is similar to McCullough Peaks rates (Abels et al., 2016). Based on observed differences in sedimentation rates, moving from northwest to southeast in the basin, we expect rates to be slower in the lower part of our section. Our ETM2 and H2 CIEs are separated by ~22 m in both sections where they are measured together and these hyperthermals are thought to be orbitally paced by 100 kyr eccentricity cycles (Barnet et al., 2019). This age constraint provides a depositional rate of ~4.5 kyr/m in the lower part of our section. Based on eccentricity maxima, H2 and I1 are separated by ~300 kyr (Barnet et al., 2019). Assuming this slower sedimentation rate, we would expect to see ~67 m of strata between H2 and I1, which is approximately 3x what we observe (Fig. 6). If we assume depositional rates increase up section toward Red

Butte, expected thickness would be even greater. Notably, there is a fairly large discrepancy in stratigraphic thickness between CIE2 and CIE3 in our composite section, measured with a Jacob's Staff and bed traces, compared with thicknesses estimated from BCM levels and dGPS elevation. Elevation and BCM levels suggest there may be as much as twenty additional meters separating H2 from the next highest excursion in our section (Fig. 4). These excursions were measured in separate sections (North Fork and Upper North Fork) tied together with a long bed trace of 1.6 km (Fig. A4 and A5), which could result in some error. Another source of error could be differences in placement of stratigraphic levels within fossil localities. BCM levels for localities are based on the stratigraphic position of a key fossil bearing bed within a locality, but many localities have several fossil bearing horizons. In most cases, we were uncertain which level was used by Bown et al. (1994) to demarcate a locality's stratigraphic level. Although we are unable to provide a confident reason for this stratigraphic discrepancy, the addition of 20 meters to the distance between H2 and the next highest CIE still falls far short of the expected thickness if the latter represents I1. Better age control is needed for a confident identification CIE3.

## 5.3 Comparison to other post-PETM hyperthermal records

The carbon isotope values vary between Fifteenmile Creek and McCullough Peaks (Fig. 6). In McCullough Peaks, the average magnitude of the ETM2 and H2 CIEs are 3.8 ‰ and 2.8 ‰, respectively. In Fifteenmile Creek, these magnitudes are 3.1 ‰ and 1.7 ‰. The smaller CIE magnitudes in Fifteenmile Creek indicate that these atmospheric perturbations are recorded differently in different parts of the same basin and likely reflect local differences in vegetation structure or soil moisture. The strongest single environmental control on $\delta^{13}C$ values in C3 vegetation is water availability to plants, which accounts for about half of the variability, with plants discriminating less against $^{13}C$ under drier conditions (Ehleringer, 1993; Stewart et al., 1995; Diefendorf et al., 2010; Kohn, 2010, 2016). This suggests that drainage differences between McCullough Peaks and Fifteenmile Creek could contribute to some of the variation between the two records. Abels et al. (2016) found that precipitation change during the post-PETM hyperthermals appears to have been negligible, based on mean annual precipitation estimates using the CALMAG proxy. This contrasts with drier conditions interpreted during the PETM from various soil proxies and fossil leaves (Wing et al., 2005; Kraus et al., 2013; Abels et al., 2016). Paleosol analysis across the Bighorn Basin shows that soil drainage varies significantly even across a small geographic area (e.g., Kraus, 1992, 1997). Secord et al. (2008) inferred vegetation structure based on carbon isotopes in mammalian tooth enamel from the Fifteenmile Creek area and found that vegetation was relatively dense, but that the forests had an open canopy. The "canopy effect," resulting from highly $^{13}C$-depleted leaves in the understory (e.g., Cerling et al., 2004), also appears to be absent in mammalian enamel from the northern Bighorn Basin (Koch et al., 1995) based on the model in Secord et al. (2008). Given the strong control that water availability has on leaf values, this appears to be the most likely cause of differences in absolute $\delta^{13}C$ values and in differences in CIE magnitude observed here, although differences in floral composition between the two locations cannot be ruled out as contributing to these differences. The slower sediment accumulation in Fifteenmile Creek also means that pedogenic carbonate in this area may have averaged atmospheric carbon isotope values over a longer time period and thus not recorded the minimum excursion values as reliably if minimum atmospheric values lasted only a short time.

Despite having lower magnitudes than the CIEs in McCullough Peaks, the Fifteenmile Creek CIEs are still amplified relative
to the marine benthic record, where the average ETM2 and H2 CIEs are ~1.4 ‰ and 0.8 ‰, respectively (Stap et al., 2010).
The amplification effect, however, seems to be somewhat variable even in similar stratigraphic records that are only ~80 km
apart within the same basin. This supports the hypothesis that some (or all) of the observed difference in magnitudes between
marine and terrestrial CIEs is due to local environmental or time averaging effects.

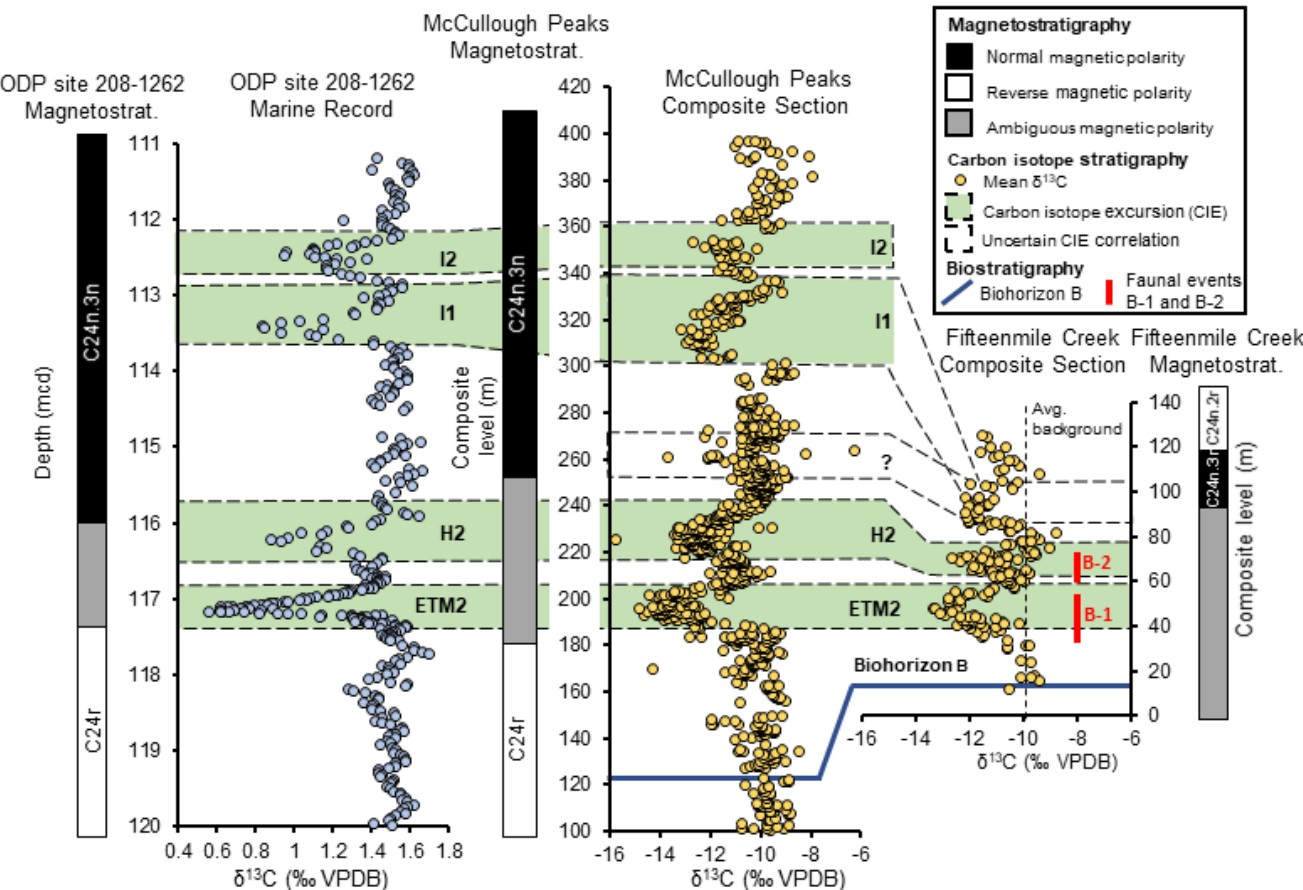

**Figure 6: Correlation between ODP site 208-1262 (from Zachos et al., 2010), McCullough Peaks (from Abels et al., 2016) and Fifteenmile Creek (this study). All points represent mean δ¹³C values from a stratigraphic level. McCullough Peaks and Fifteenmile Creek records are shown at the same vertical scale. Two alternative correlations for the 85 – 104 m Fifteenmile Creek composite level interval are shown. Blue line shows the approximate level of Biohorizon B in both terrestrial locations and red bars show the**
**levels associated with faunal events B-1 and B-2 in Fifteenmile Creek. McCullough Peaks magnetostratigraphy from Clyde et al. (2007), Abels et al. (2012), and D'Ambrosia et al. (2017). Fifteenmile Creek magnetostratigraphy from Clyde et al. (2007) (measured at Elk Creek Rim) and Tauxe et al. (1994).**

## 5.4 Correlation to mammal faunal events

The placement of the ETM2 and H2 CIEs in our new sections aligns with the levels predicted by Chew (2015) based on the levels of the B–1 and B–2 faunal events (Figs. 4 and 5). Error in the stratigraphic correlations used to construct the BCM framework or averaging of specific collecting levels within localities could both contribute to the small offsets between the CIE levels and the BCM levels that Chew (2015) predicted. Many of these localities have a relatively small number of specimens and would therefore have little effect on the faunal analysis (e.g., D-1403 has 34 specimens). The misplacement of richer localities, like D-1454 (1151 specimens) and D-1460 (405 specimens), would be more significant. These minor inconsistencies have likely muted the turnover signal observed by Chew (2015). Moreover, a confounding problem with correlating faunal turnover to many of these localities, is that fossiliferous horizons often occur at multiple levels within a locality. For example, in locality D-1350, mammal fossils have been collected both above and below the onset of the ETM2 excursion and most collection records are not precise enough to indicate which specimens came from which levels. Thus, a turnover signal could be muted by mixing taxa from within the excursion with pre- and post-excursion taxa. This problem can be overcome only by recollecting such localities using higher stratigraphic precision.

More importantly, these results demonstrate that there is a reasonably large miscorrelation in the BCM levels of the localities south of Fifteenmile Creek, in the Basin Draw section, compared with north of the creek. South of Fifteenmile Creek, the BCM levels alone are not an accurate predictor of a locality's position relative to the hyperthermals. For example, the D-1454 and D-1460 localities (409 BCM level) are not within a CIE, despite this BCM level occurring within faunal event B–1 and overlapping with ETM2 in other sections (e.g., Basal/D-1350). We sampled a smaller isotope section directly through the D-1454 locality to test whether incorrect correlation to the main section could have contributed to this offset or in fact it falls within a CIE. The $\delta^{13}C$ record through the locality is consistent with that from the main section and suggests that it is not within ETM2 (Appendix B). It seems likely that the Basin Draw section localities should be shifted downwards by ~20 meters in the Bown et al. (1994) composite section. This would align the excursion at the top of Basin Draw with CIE1 and place the productive localities D-1454 and D-1460 within Biohorizon B. Alternatively, shifting the section up by ~25 meters would align the low isotope values near the base of the Basin Draw section with CIE1 and place the productive localities D-1454 and D-1460 closer to faunal event B–2. The biostratigraphy is equivocal in terms of favoring one adjustment over the other, although the dGPS elevation supports the first option. Making one of these stratigraphic adjustments would likely strengthen the distinction between Biohorizon B and the subsequent faunal events B–1 and B–2 in a revised faunal analysis should further stratigraphic work demonstrate that this move is warranted. The possibility of a buried fault along Fifteenmile Creek could be responsible for some of the discrepancies in correlation across the creek. This would also be consistent with other observations of east-west trending fault activity during Eocene deposition (Kraus, 1992).

The isotope results presented here suggest that correlation of key localities should be tested by constructing new locality-specific isotope records that can be directly tied to the mammal collection from the same locality, similar to the sections through D-1454, D-1207, and D-1532 (Appendix B). For example, the locality-specific isotope record for locality D-1207

(448 BCM level) shows that this locality records the onset of the H2 CIE, rather than its recovery, as predicted by the BCM levels. Developing similar sections through other individual localities, where the primary fossil producing layer can be reliably constrained and tied into a local dGPS elevation and stable isotope record and then compared to other localities either within or outside of CIEs, would clarify observed faunal turnover and its potential relationship to climate fluctuations. Such precise correlations could also be used to investigate other stratigraphic levels where we find CIEs, such as the 486 m BCM level, to see whether there is associated faunal change.

## 6    Conclusions

The ETM2, H2 and potentially I1 hyperthermals are recognized for the first time in pedogenic carbonate from the south-central Bighorn Basin (Fifteenmile Creek area). Correlation of the ETM2 and H2 carbon isotope excursions with the faunal record supports previous suggestions that some mammalian turnover during the early Wasatchian North American Land Mammal age, following the PETM, is associated with early Eocene hyperthermals. The new isotope records presented here also highlight some of the complexities of lithological correlation between the low-lying exposures in the Fifteenmile Creek area and support the use of an independent means of stratigraphic correlation, such as precise elevation, in conjunction with mammalian biostratigraphy and carbon isotope chemostratigraphy. The magnitudes of the ETM2 and H2 CIEs (3.1 ‰ and 1.7 ‰, respectively) are smaller than what is seen farther north in the McCullough Peaks region of the Bighorn Basin (3.8 ‰ and 2.8 ‰). We suggest that local variation in water availability to plants, and potentially other vegetation and soil processes, likely account for much of the differences in carbon isotope values observed between the two locations.

## Appendix A

To sample in the relatively low relief terrain of the Fifteenmile Creek area, the six main sections were measured as a series of subsections (Figs. A1 – A6). Subsections were measured using a Jacob's staff and tied together either by tracing the base or top of a marker bed between subsections or correlation by siting a horizontal line (Fig. A7). Differential GPS elevations were measured from each section to place the carbon isotope results into an elevation framework but were not used to correlate between sub-sections. Fossil localities were either sampled directly or traced into the sections to allow the isotope data to be placed into the composite stratigraphy of Bown et al. (1994) (Fig. A8). Meter levels within an individual subsection and bed traces over short distances have less than 1 m of uncertainty, typical for measurements using a Jacob's staff and clinometer. In some places, sections were correlated to one another using bed traces and in others, correlations were made using BCM levels for fossil localities that were measured into local sections (Fig. A9). Bed traces over longer distances between sections may have an uncertainty of up to ~10 meters.

Sampling was done over four years (2015 – 2018) and differential GPS elevation measurements were taken during the final year of field work (2018). Differential GPS points were preferentially taken at levels that correspond to isotope excursions and

secondarily at the base and top of each subsection. In places where the base/top of a subsection was uncertain at the time of dGPS measurements, a sample site was selected that could be confidently located to help constrain the subsection. Carbon isotope results for each section are included in Fig. A10, with results from different subsections indicated.

**Basin Draw:** The Basin Draw section is made up of 7 subsections and includes ~60 total meters of section (Fig. A1). Bed
tracing over a long distance was facilitated by the presence of a distinct red – purple – white "barren marker bed". Three fossil localities (D-1458, D-1454, D-1460) were correlated to the section via bed traces and a fourth locality (D-1459) was sampled directly for isotopes. Locality D-1454 was also sampled independently as a shorter section (Appendix B). Subsection 6 includes the fossil locality D-1459 (438 m BCM level) and appears to be part of a cut-and-fill sequence, introducing some additional uncertainty into the upper part of this section. It was not possible to trace beds between the Basin Draw section and other
sections in this study due to the intervening Fifteenmile Creek. Correlation between Basin Draw and the other sections therefore relies on the stratigraphic level of fossil localities, which can be supplemented with dGPS elevations and chemostratigraphic patterns.

**Basal/D-1350:** The Basal/D-1350 section includes 6 subsections and ~52 total meters of section (Fig. A2). In places, individual samples were collected along a bed trace and were not part of a separate subsection (e.g., between subsections 1 and 2 and
between subsections 4 and 5). Correlation between subsections 2 and 3 and between subsections 3 and 4 were done by siting a horizontal line between exposures and these correlations have an uncertainty of $\pm$ 1.5 m and $\pm$ 5 m, respectively. Fossil localities D-1370 and D-1758 were correlated into the section via bed traces and localities D-1350 and D-1538 were sampled directly. Lining up minimum CIE values from the top of this section and the base of the North Fork section offers the most reliable means of correlating the two sections. The 409 BCM level at the D-1350 locality can also be used to correlate to the
409 BCM level D-1454 and D-1460 localities in the Basin Draw section.

**Kraus Flats:** The Kraus Flats section was measured as a parallel section to North Fork to capture the middle two CIEs in a section with additional fossil localities. This section allowed the productive fossil locality D-1204 ("Kraus Flats") to be tied into the isotope framework. It includes 5 subsections and ~ 55 total meters of section (Fig. A3). Fossil locality D-1410 was sampled directly through two distinct fossil producing levels (D-1410 mid and D-1410 top). D-1410 mid also preserves the
H2 CIE in a yellow-grey mottled paleosol (Fig. A7). Localities D-1822 and D-1204 were correlated to the section via bed traces. A distinct purple marker bed ("Kraus Flats purple marker") facilitated correlation for the top of the section, moving up towards locality D-1204. No beds were traced between Kraus Flats and the other sections. Correlating minimum CIE values is the preferred method for tying Kraus Flats to the other sections.

**North Fork:** The North Fork section includes 8 subsections and ~35 total meters of section (Fig. A4). Two distinct and
relatively continuous purple marker beds ("Purple #1" and "Purple #2") facilitated correlation between several short subsections. Subsections 4 and 5 were sampled at a later date to fill in sampling gaps around potential CIE levels. These subsections were tied into the existing North Fork framework by measuring up to existing sample sites that could be confidently located (CBH15-SN32 and CBH15-SN33). Fossil locality D-1403 was correlated to subsection 1 via bed tracing and fossil localities D-1295 and D-1311 were sampled directly. A purple bed near the top of subsection 6 was traced to the base of the

Upper North Fork section. The long distance of this trace, combined with evidence from dGPS elevations and BCM levels for the two sections suggest an uncertainty of ~10 m on this correlation.

**Upper North Fork:** The Upper North Fork section includes 3 subsections and ~28 total meters of section (Fig. A5). The highest identified CIE (CIE3) is preserved in a yellow-grey paleosol, just above a thin, discontinuous grey sandstone (Fig. A7). Fossil locality D-1257 was correlated to subsection 3 in this section via bed trace. The base of a purple bed in a red-purple

couplet was traced between subsections 1 and 2 and also between subsection 2 and the base of our Red Butte section. This bed trace has ~10 m of uncertainty, due to the long distance.

**Red Butte:** The Red Butte section includes 3 subsections and ~48 total meters of section (Fig. A6). The purple bed near the base of subsection 1 was traced from the Upper North Fork section. Isotope samples near the base of subsection 1 were collected just above fossil locality D-1491. Fossil locality D-1532 is located near the Red Butte section and was correlated to

the red-purple couplet at the base of this section as well. D-1532 was also sampled independently as one of the short fossil locality isotope sections (Appendix B).

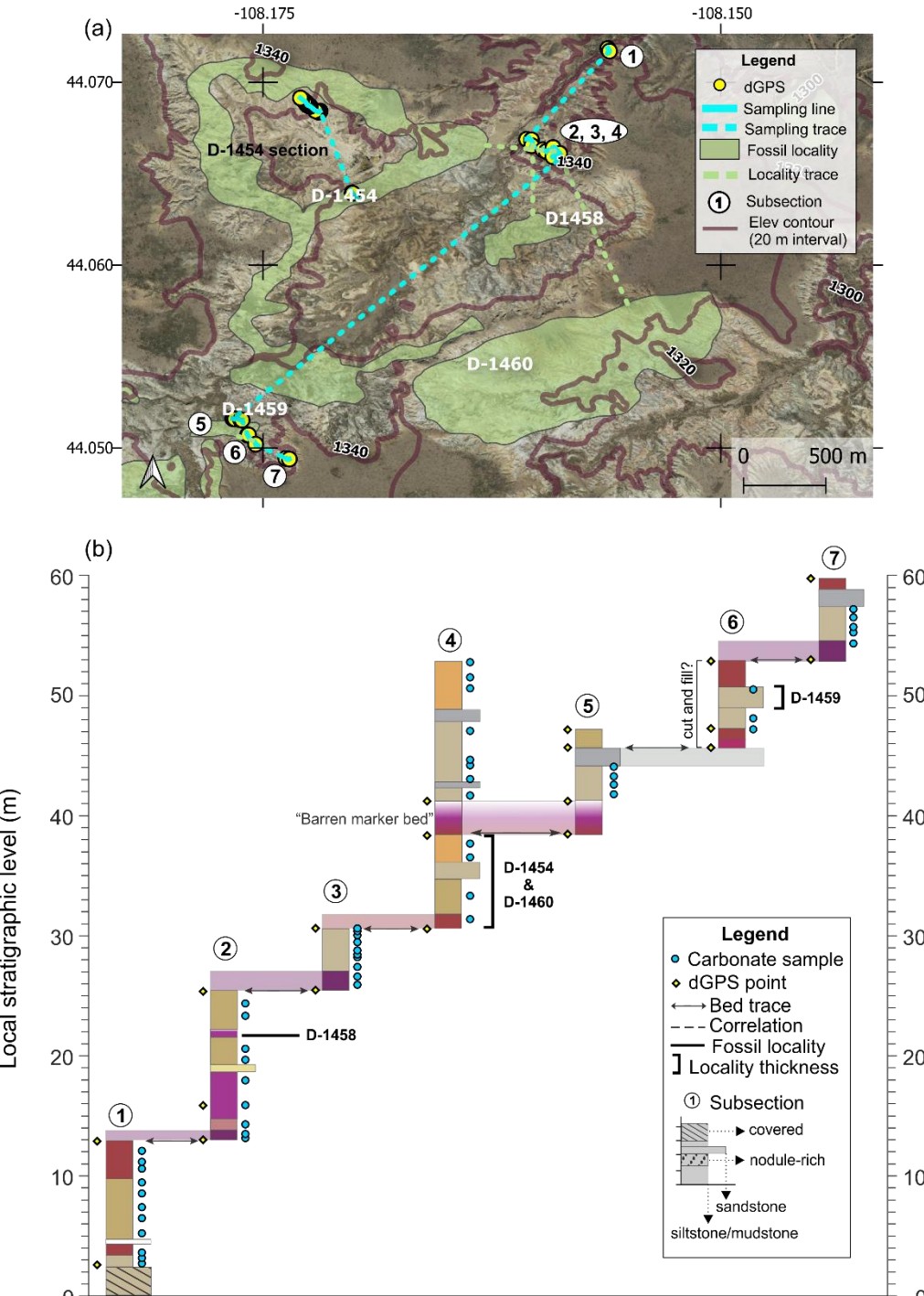

**Figure A1: Basin Draw section (a) map and (b) diagram showing sampling subsections, dGPS tie points, bed traces, and correlated fossil localities. Colored boxes in (b) represent the surface color of weathered rocks. Marker beds used to trace between subsections are shown in light shading with arrows along the bed surface used for tracing. Basemap source: Google Earth © 2021. A high-resolution map is available in the supplemental data repository.**

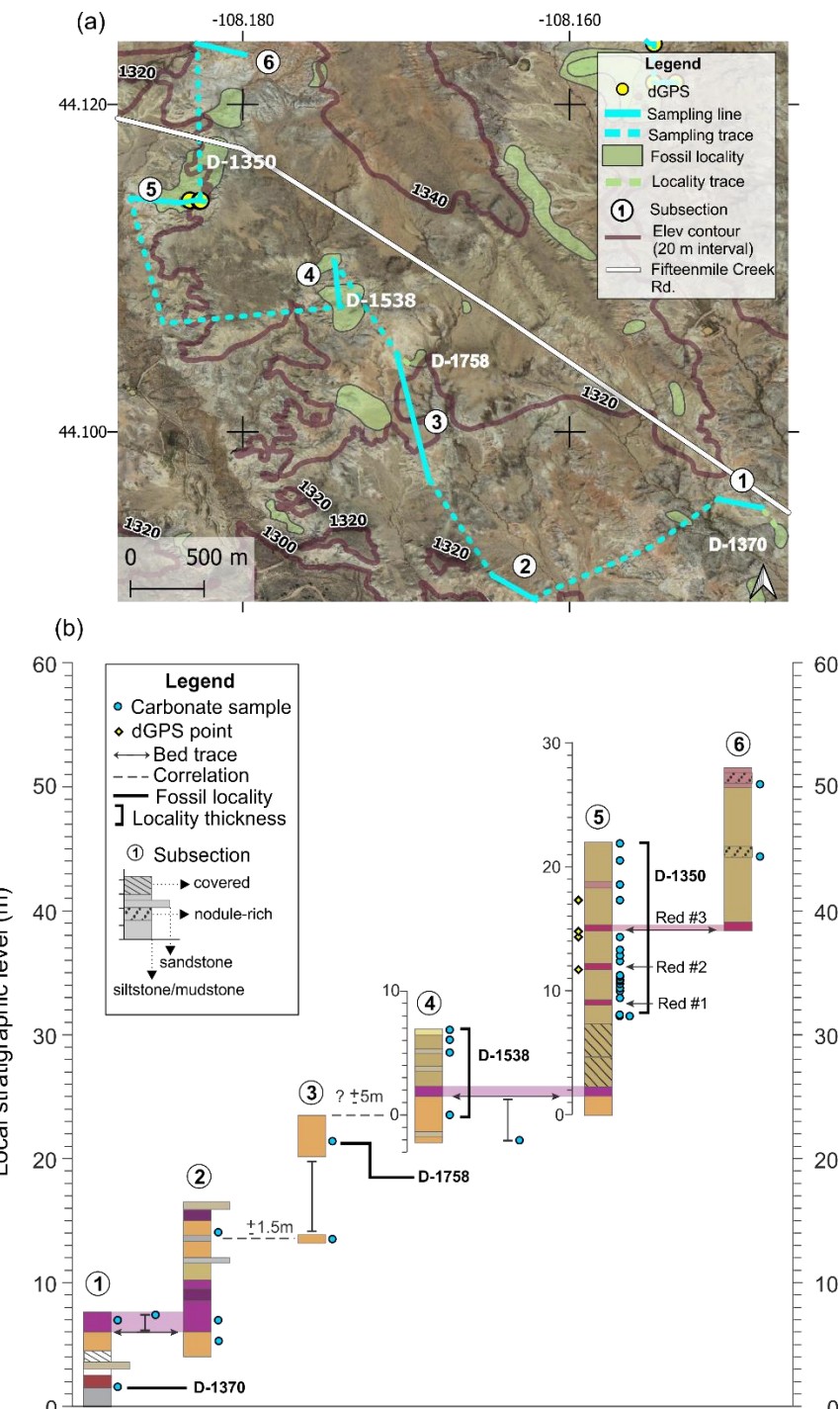

**Figure A2:** Basal/D-1350 section (a) map and (b) diagram showing sampling subsections, dGPS tie points, bed traces, and correlated fossil localities. Colored boxes in (b) represent the surface color of weathered rocks. Marker beds used to trace between subsections are shown in light shading with arrows along the bed surface used for tracing. Basemap source: Google Earth © 2021. A high-resolution map is available in the supplemental data repository.

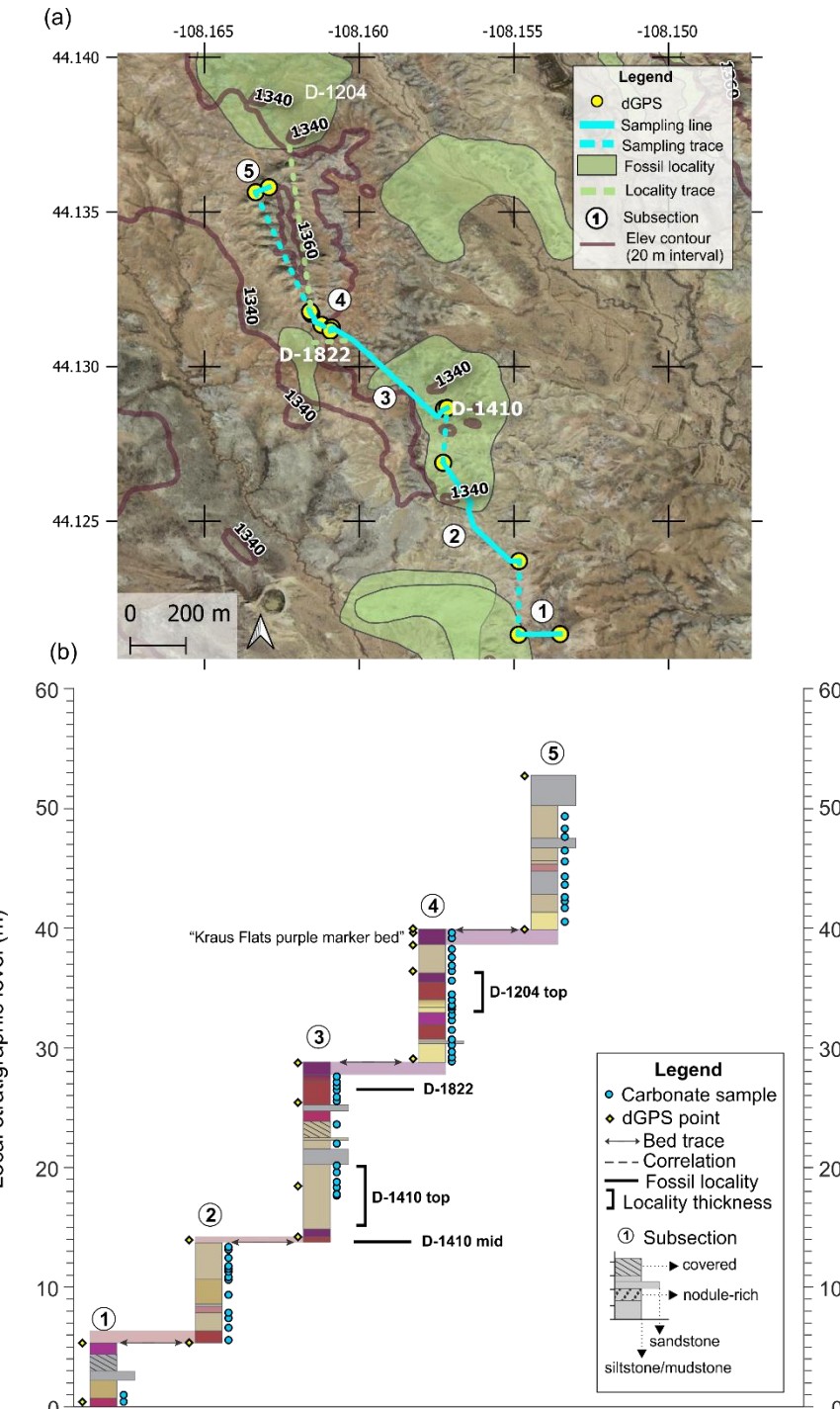

**Figure A3:** Kraus Flats section (a) map and (b) diagram showing sampling subsections, dGPS tie points, bed traces, and correlated fossil localities. Colored boxes in (b) represent the surface color of weathered rocks. Marker beds used to trace between subsections are shown in light shading with arrows along the bed surface used for tracing. Basemap source: Google Earth © 2021. A high-resolution map is available in the supplemental data repository.

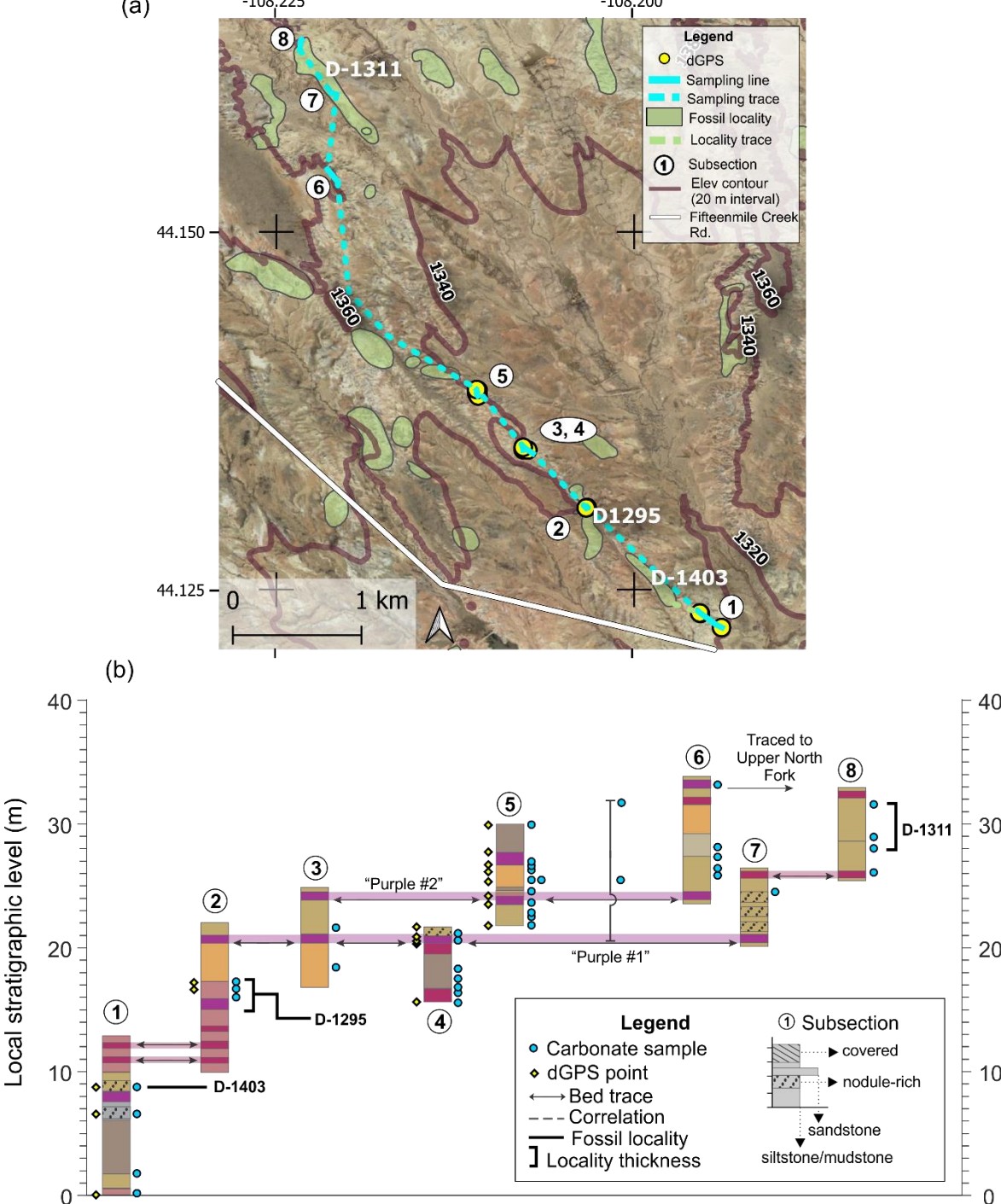

**Figure A4: North Fork section (a) map and (b) diagram showing sampling subsections, dGPS tie points, bed traces, and correlated fossil localities. Colored boxes in (b) represent the surface color of weathered rocks. Marker beds used to trace between subsections are shown in light shading with arrows along the bed surface used for tracing. Basemap source: Google Earth © 2021. A high-resolution map is available in the supplemental data repository.**


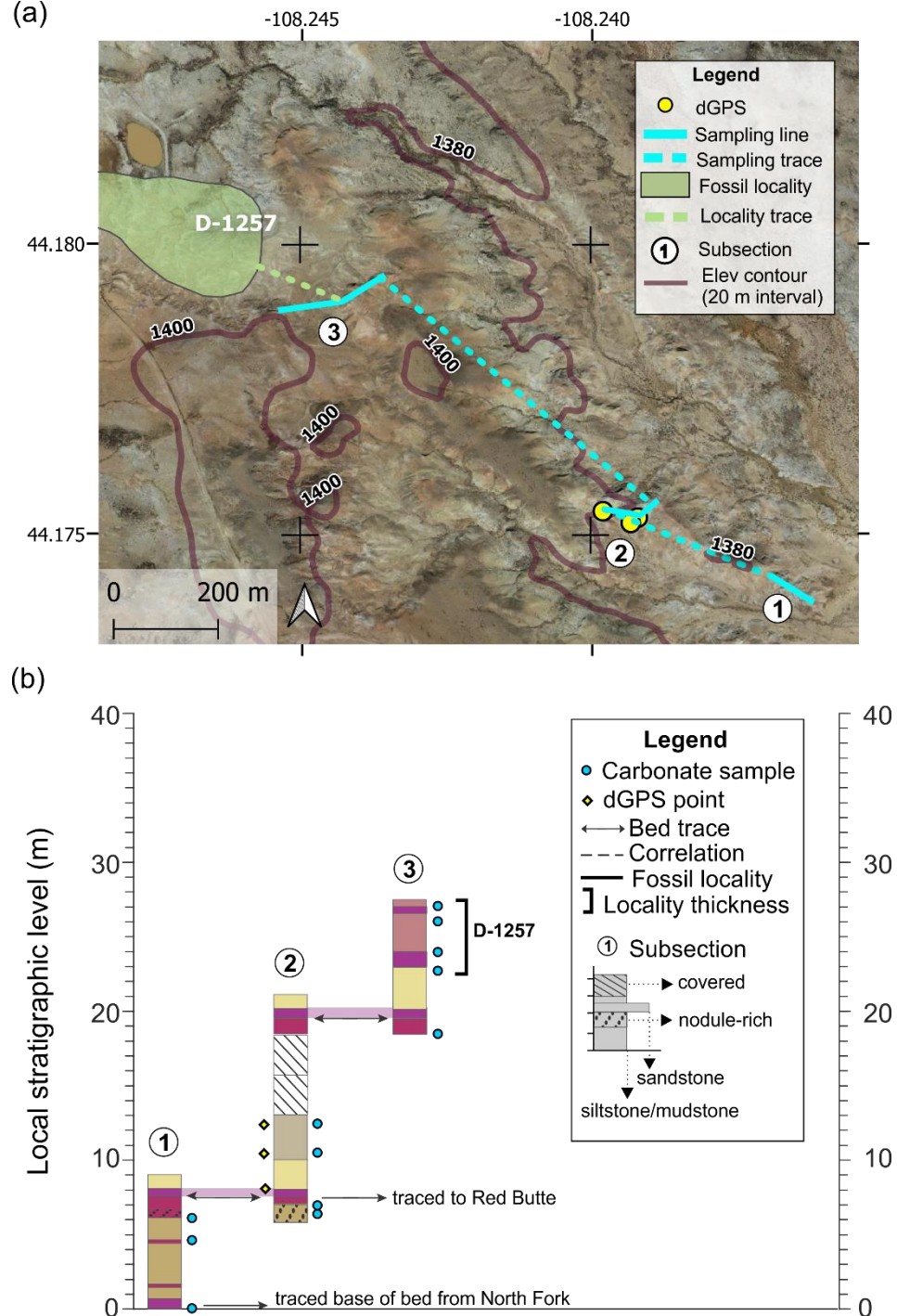

**Figure A5:** Upper North Fork section (a) map and (b) diagram showing sampling subsections, dGPS tie points, bed traces, and correlated fossil localities. Colored boxes in (b) represent the surface color of weathered rocks. Marker beds used to trace between subsections are shown in light shading with arrows along the bed surface used for tracing. Basemap source: Google Earth © 2021. A high-resolution map is available in the supplemental data repository.

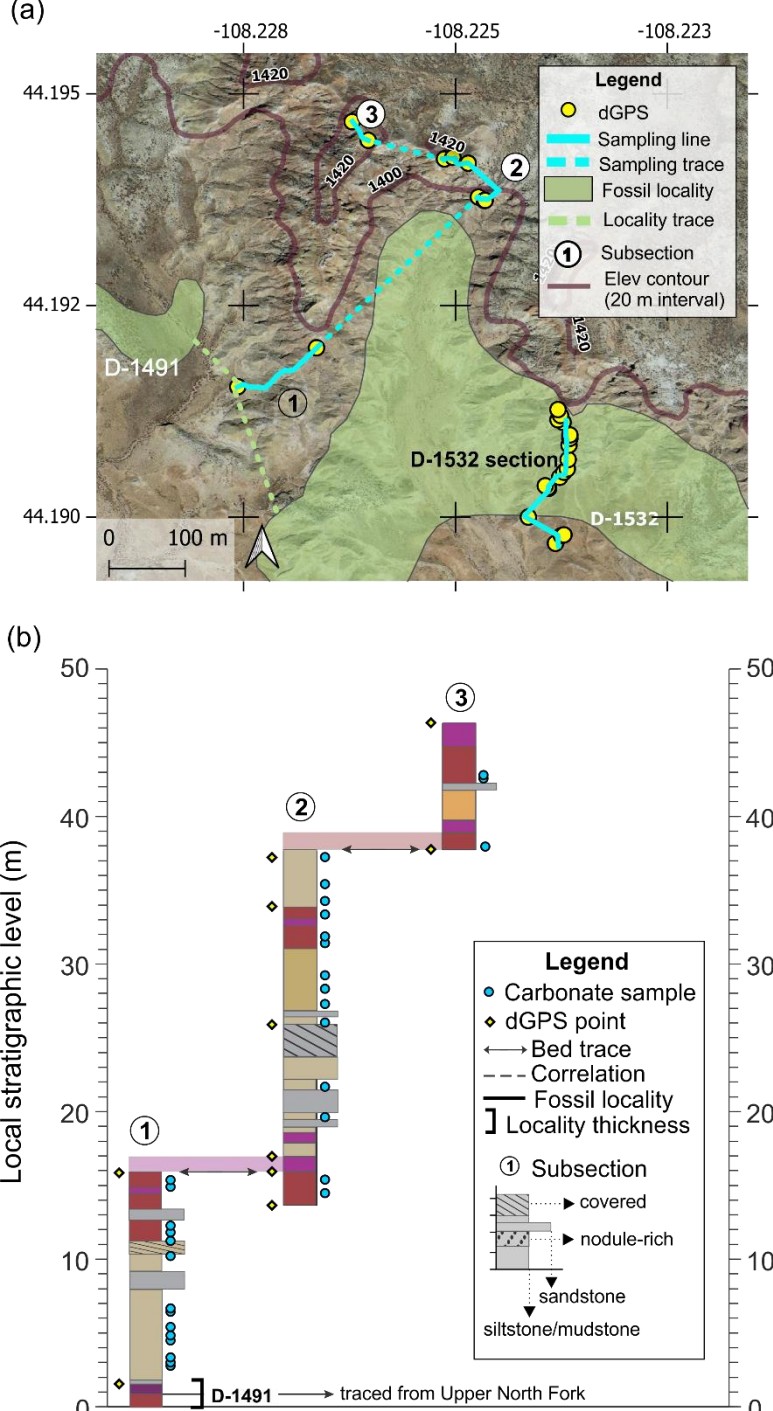

**Figure A6: Red Butte section (a) map and (b) diagram showing sampling subsections, dGPS tie points, bed traces, and correlated fossil localities. Colored boxes in (b) represent the surface color of weathered rocks. Marker beds used to trace between subsections are shown in light shading with arrows along the bed surface used for tracing. Basemap source: Google Earth © 2021. A high-resolution map is available in the supplemental data repository.**


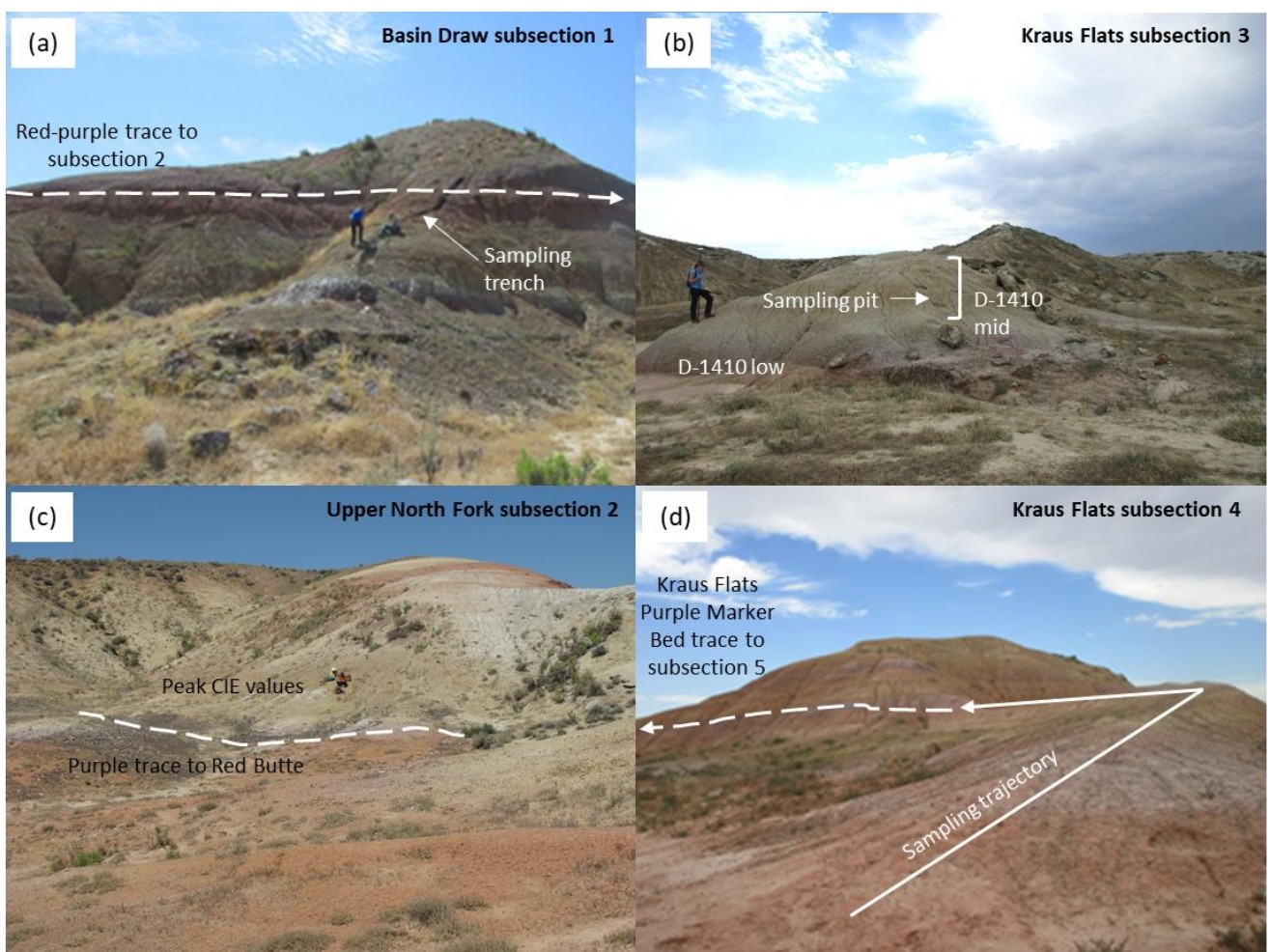

**Figure A7: Field photos of representative sections. (a) Basin Draw subsection 1, showing sampling trench and red-purple contact traced to subsection 2. (b) Kraus Flats Subsection 3, showing fossil locality D-1410 collecting levels and carbonate sampling pits. Yellow bed with sample pits represents middle collecting level in D-1410 locality and ETM2 excursion. (c) Upper North Fork subsection 2, showing red-purple contact traced to Red Butte section and peak CIE3 excursion in the yellow paleosol. (d) Kraus Flats subsection 4, showing sampling trajectory through the H2 excursion and Kraus Flats Purple Marker Bed traced to subsection 5. Person for scale in a – c. High resolution photos are available in the supplemental data repository.**


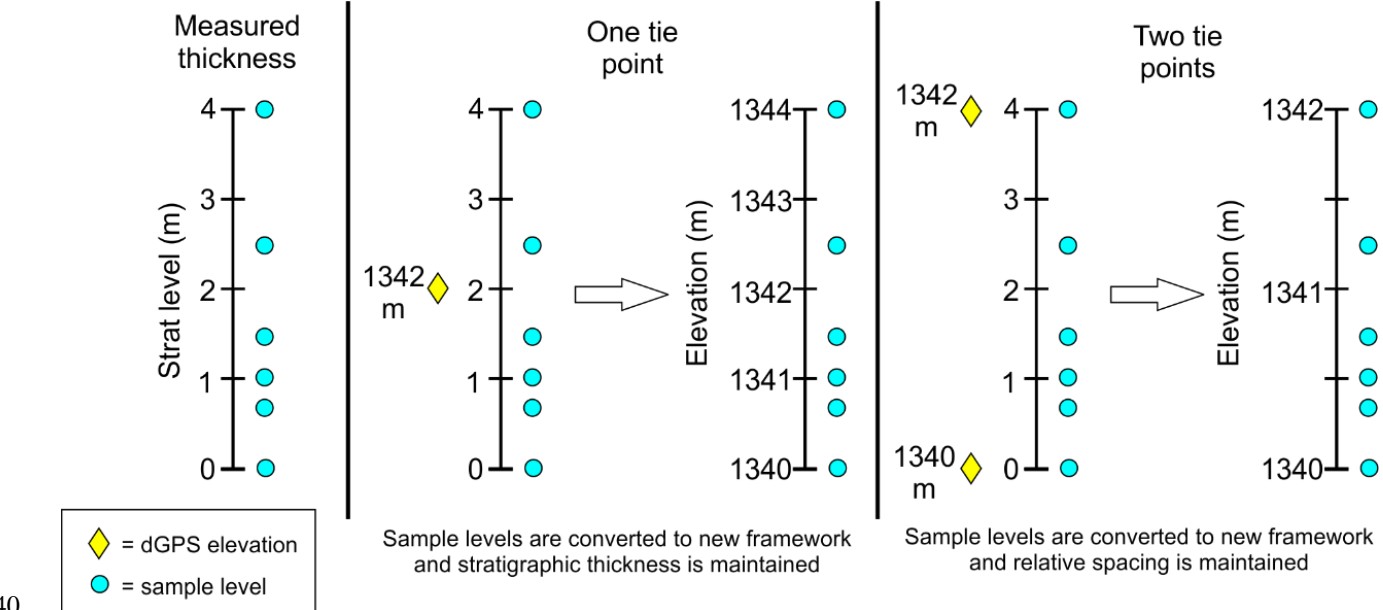

**Figure A8: Conversion between measured stratigraphic thickness to elevation using one or two tie points. The same method was used for converting to BCM levels using correlated fossil localities as tie points.**

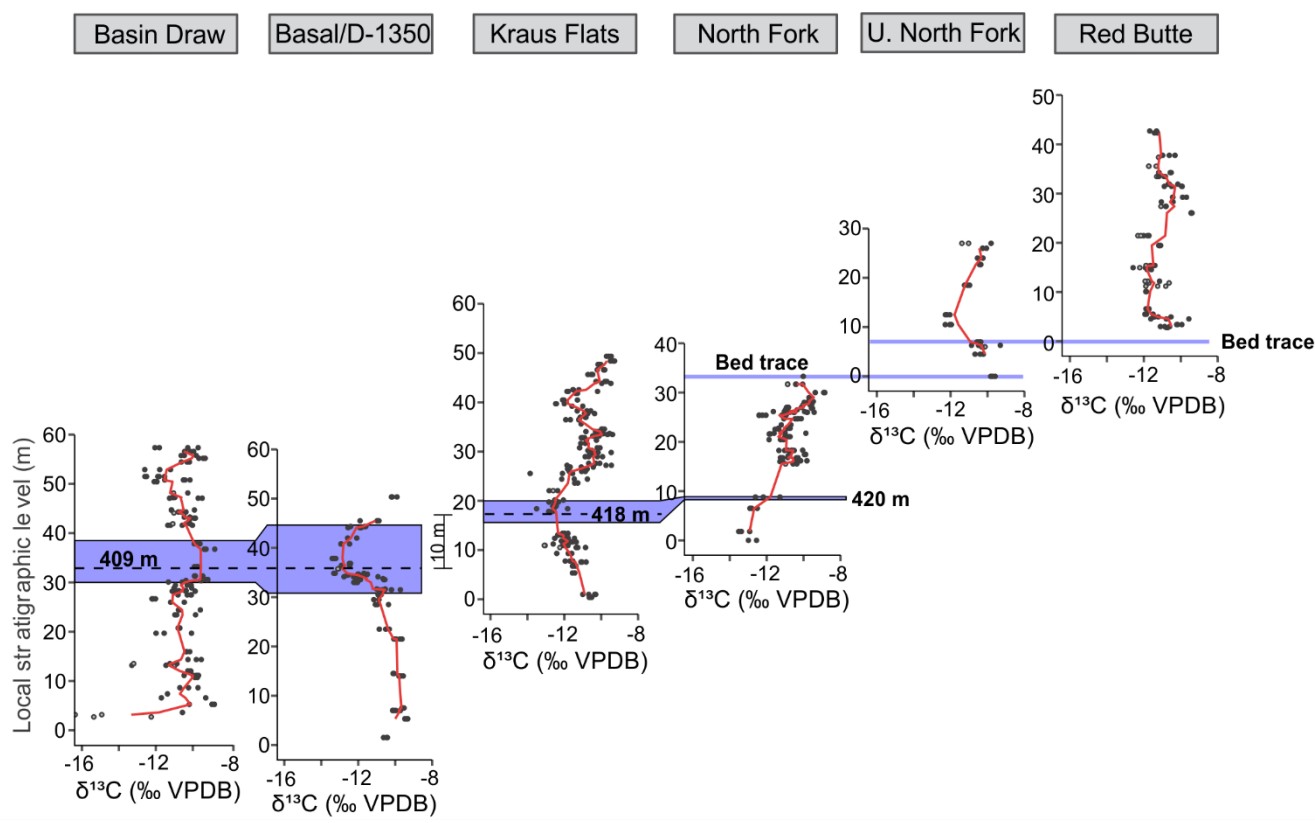


**Figure A9: Section correlations based on bed traces and fossil locality levels. Stratigraphic levels represent meter levels measured using a Jacob's staff. Bed traces are used to correlate between the North Fork and Upper North Fork sections and between the Upper North Fork and Red Butte sections. Fossil localities that occur at the 409 BCM level in the Basin Draw and Basal/D-1350 sections are used to correlate these sections. Fossil localities that occur near the 420 BCM level in the Kraus Flats and North Fork**
**sections are used to correlate these sections and the difference between these BCM levels helps tie the Basin Draw, Basal/D-1350, and Kraus Flats sections together.**

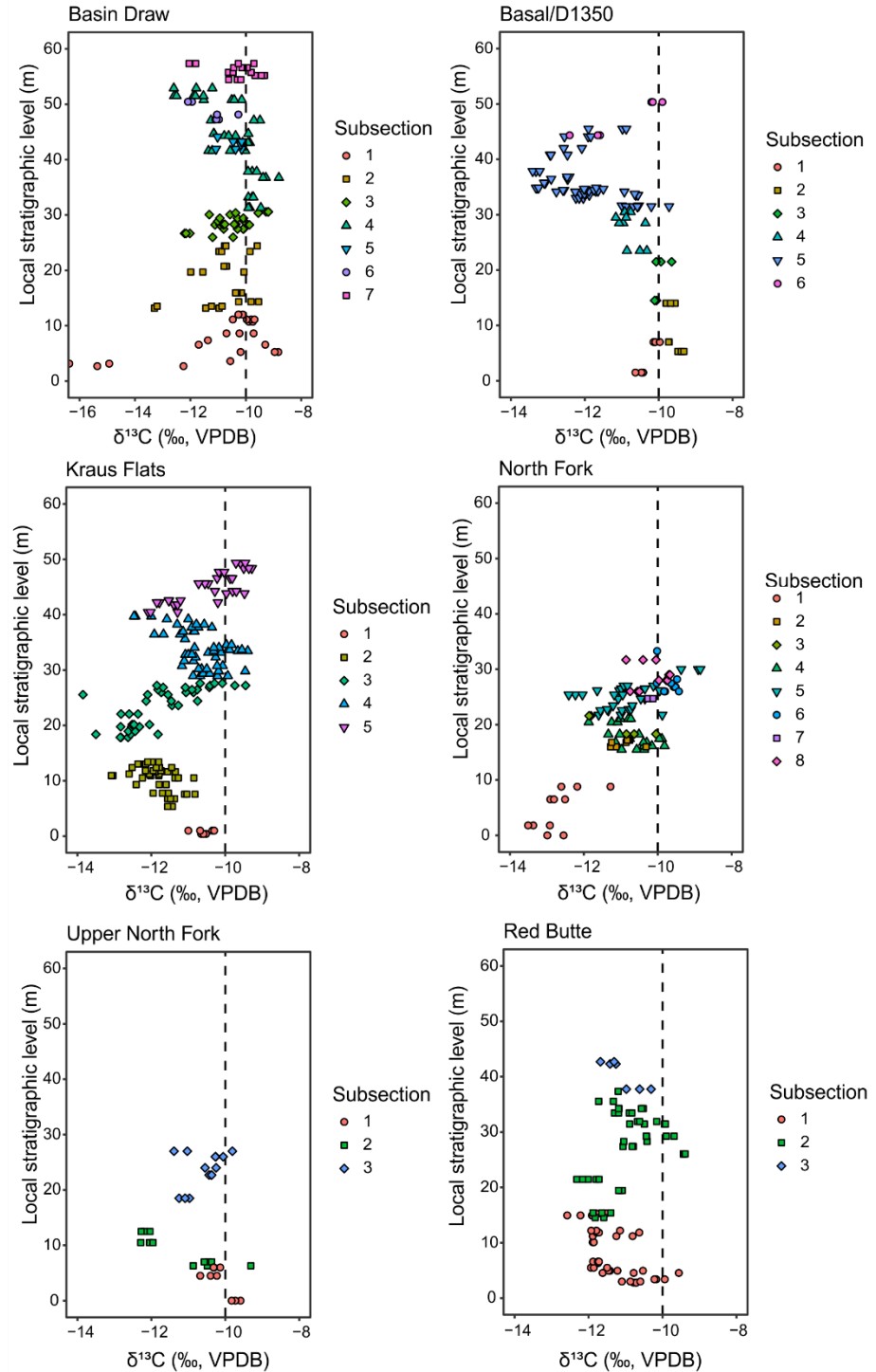

**Figure A10: Carbon isotope stratigraphy for each section with symbols showing different subsections. Points represent individual sample measurements. Black dashed line at -10 ‰ shows typical background value for Fifteenmile Creek.**

**Appendix B**

**Carbon isotopes stratigraphy through fossil mammal localities**

Three locality-specific stratigraphic sections were sampled for stable isotope analysis to identify whether they captured a CIE. If a CIE fell entirely within the fossil producing levels of a single locality, future fossil collecting could target these areas for more refined faunal analyses that would eliminate uncertainty related to stratigraphic averaging and correlation between sections. The three localities: D-1454 (409 BCM), D-1207 (448 BCM), and D-1532 (485 BCM) were selected because they are found at important levels associated with faunal events B–1 (D-1454) or B–2 (D-1207) or near an unresolved CIE (D-1532). These smaller locality-specific sections can then be compared to their nearest main section using the BCM levels and elevation to further assess spatial variability in $\delta^{13}C$ and confirm the correlations. The carbon isotope values are relatively consistent over short distances in Fifteenmile Creek (Fig. B1). The Bown et al (1994) composite level (BCM) and dGPS elevation are also both fairly reliable stratigraphic indicators over short traces and correlate the locality sections to the main sections in consistent ways (Fig. B1). An exception to this is correlating the D-1207 locality to the Kraus Flats section, where the BCM level lines up with the H2 excursion, but the dGPS elevations have D-1207 and H2 offset from each other. D-1207 (448 BCM level) is the only one of these three localities that falls within an excursion (H2). This level is predicted to be within the B–2 faunal event in Chew (2015) (435 to 448 m). D-1207 has two taxa that first appear in faunal event B–2 (*Hexacodus sp.* and *Protorohippus venticolum*), and this locality therefore represents a place where the carbon isotope stratigraphy and biostratigraphy are consistent and suggest that it falls within H2 and the B–2 faunal event. It also illustrates a need to use both BCM and dGPS for precise correlation of carbon isotope stratigraphy in this area.

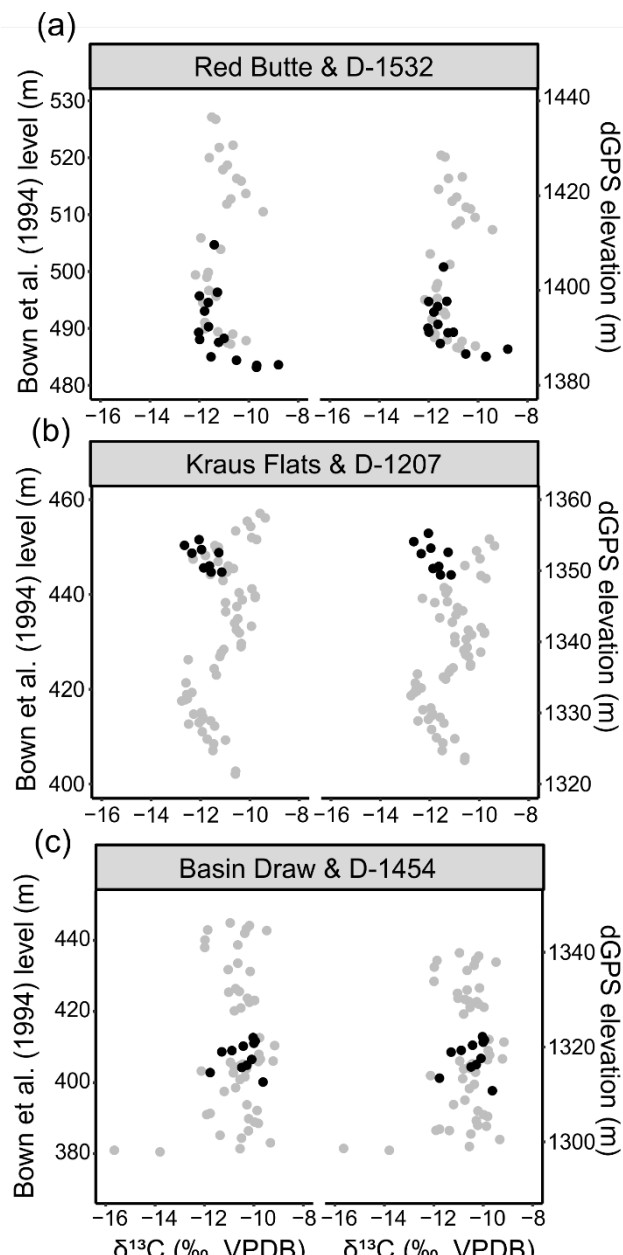

Figure B1: Carbon isotope stratigraphy through the (a) D-1532 (485 BCM level), (b) D-1207 (448 BCM level), and (c) D-1454 (409 BCM level) fossil localities (black) along with the nearest main section (grey). Points represent average δ¹³C from a stratigraphic level. All data are shown according to their Bown et al. (1994) equivalent meter level (left) and differential GPS elevation (right).

**Appendix C**

**Inter-laboratory offsets and correction**

Comparison of the carbon and oxygen isotope data from samples run at both CUBES–SIL and UMSIL shows that CUBES–SIL produces generally lower values than UMSIL in this dataset's range of values. This offset to lower values is systematic and is larger at progressively lower values. This issue affects both carbon and oxygen isotopes but affects carbon more, which

suggests that subtle differences in instrumentation and standardization procedure are the causes of this offset, as discussed below. UMSIL uses a ThermoFisher MAT253 attached to a Kiel IV automated preparation device and data are normalized using a best-fit linear regression to NBS-18 and NBS-19 reference materials ($\delta^{13}C$ = -5.01 ‰ and +1.95 ‰, $\delta^{18}O$ = -23.2 ‰ and -2.20 ‰, respectively) by regressing measured, $^{17}O$-corrected values against the published values for the standards. Corrections are determined for both carbon and oxygen and these corrections are checked routinely using either NBS-18, NBS-

19, or both, and tend to be stable over several months.

CUBES–SIL uses a Thermo Delta V gas source, continuous flow isotope ratio mass spectrometer. Samples are measured along with 3–4 standards that either have internationally accepted values or are in-house standards that have values determined relative to these standards (e.g., NBS-18, NBS-19, and LVSEC for light $^{13}C$ standards). Measurements of the standards bracket sample measurements in each run and are used to assess behaviors of the instrument that may need to be corrected. Standards

covering the full range of signal intensities observed in the samples are measured to assess effects of linearity. Standards are also run intermittently throughout the analytical session to evaluate instrument drift over the course of a run. Lastly, the overall offset of the standards from accepted values is evaluated. These corrections are done independently for carbon and oxygen and are checked using a monitoring standard that is treated as an unknown. In this dataset, linearity and drift were often negligible, and corrections for these effects were only applied when needed. Raw or linearity/drift corrected values were then corrected

to final values similarly to UMSIL, by applying a regression between two standards that span a range in values. Runs 1–13 and 15 used NBS-18 as the negative anchor for both $\delta^{13}C$ and $\delta^{18}O$, while runs 14, 16, and 17 used a MERCK carbonate as the negative $\delta^{13}C$ anchor point ($\delta^{13}C$ = -35.6 ‰), to better standardize for the range of $\delta^{13}C$ values found in typical terrestrial carbonates, following recommendations in Coplen et al. (2006).

Both UMSIL and CUBES–SIL used NBS-18 as the negative $\delta^{13}C$ for most measurements. NBS-18 is ~4–10 ‰ higher than

most of the measurements in this dataset, so the linear regression that is used for the scale correction in both labs is extrapolated far beyond that lowest anchor point. As a result, uncertainties in this regression in both labs, while small, can result in differences in sample values that are larger than analytical uncertainty. If this were the cause of the difference in sample values between CUBES–SIL and UMSIL, one would expect to see correlation between the values from both labs, as well as a progressively larger offset between sample values at more negative $\delta^{13}C$ values. A strong correlation and positive slope are

seen between the isotope ratios measured at CUBES–SIL relative to UMSIL in both $\delta^{13}C$ and $\delta^{18}O$ values for a small subset of samples that were run in both labs (Fig. C1). The strength of the correlation for $\delta^{13}C$ ($R^2$ = 0.88, S = 0.37 ‰) suggests that we can correct data run from one lab to be on scale with the other lab. We compared a subset of values produced at CUBES–

SIL using either NBS-18 or the lower MERCK standard for the same sample. This showed no systematic difference in the values (Fig. C2), which suggests that CUBES–SIL data are internally consistent, despite the extrapolation of the scale

correction for many of the runs. Given that, and that the majority of this dataset was analyzed at CUBES–SIL, we have chosen to correct the UMSIL data to these values, using the regression equations for both carbon and oxygen (Fig. C1). After applying the correction, the UMSIL data become more negative, while still following the same stratigraphic patterns in $\delta^{13}C$ (Fig. C3). While we think that this correction scheme is most appropriate for these data, we recognize that it may complicate comparisons to other previously published work that do not use $\delta^{13}C$ standards that account for values in this range. Further, because the

$\delta^{13}C$ correlation between labs is strong, the magnitudes of the CIE's are not appreciably affected in the combined data after correction. Oxygen isotope values exhibit a weaker correlation between the labs ($R^2 = 0.74$), indicating that considerably more noise is added by combining the data. However, highly precise $\delta^{18}O$ values are not important for our interpretations.

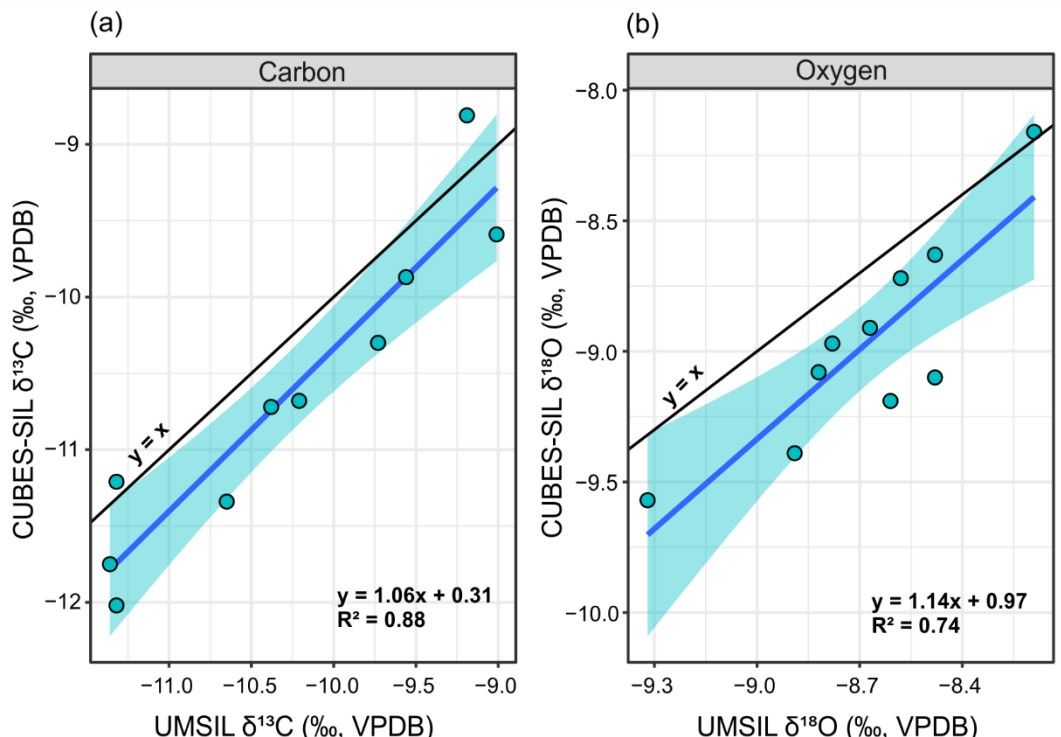

**Figure C1: Ten replicate samples analyzed at the University of Colorado Boulder (CUBES–SIL) and University of Michigan (UMSIL), shown according to their $\delta^{13}C$ (a) and $\delta^{18}O$ (b). Black lines indicate a 1:1 relationship between the two labs and blue lines with shading represent the regressions between the two labs that were used to correct the data.**

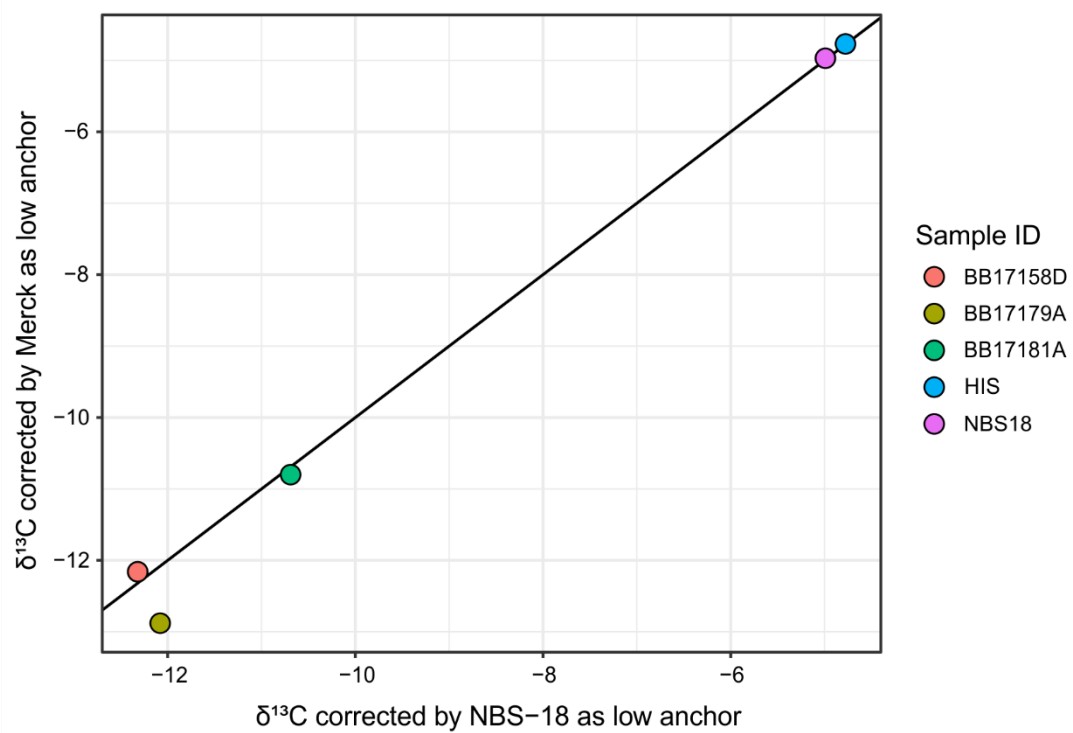

**Figure C2: δ¹³C values for standards and a subset of samples that were run in a session that used NBS-18 as a low δ¹³C anchor, as well as in a session that used MERCK as the low δ¹³C anchor. Most fall on the 1:1 line (black line), and the two samples that fall off the line show no systematic offset suggesting the difference in those values is not due to the difference in anchor standard.**

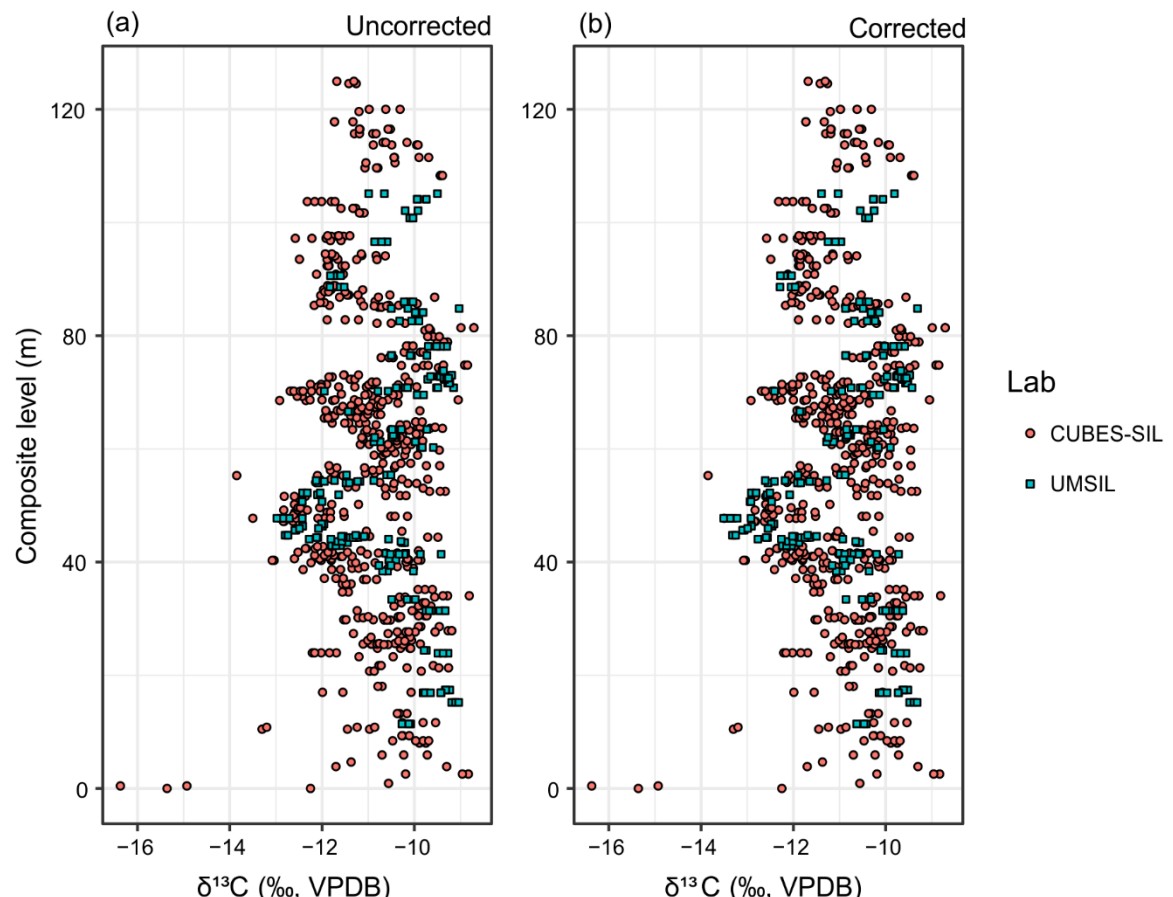

**Figure C3: Pedogenic carbonate δ¹³C showing samples that were analyzed in the UMSIL lab uncorrected (a) and corrected (b), using the carbon regression equation from Fig. B2. Points represent individual sample measurements. The corrected UMSIL values (b) are used throughout the text. This composite section includes the Basin Draw and D-1454 sections with the top of the Basin Draw section correlated to the lowest CIE (ETM2).**



**Appendix D**

**Oxygen Isotopes**

A seven-point moving average through all samples in the composite section do not show clear excursions in the oxygen isotope record (Fig. D1). Moreover, there is no discernible difference between excursion and non-excursion $\delta^{18}O$ values when these intervals are binned together and averaged (see supplemental data repository).


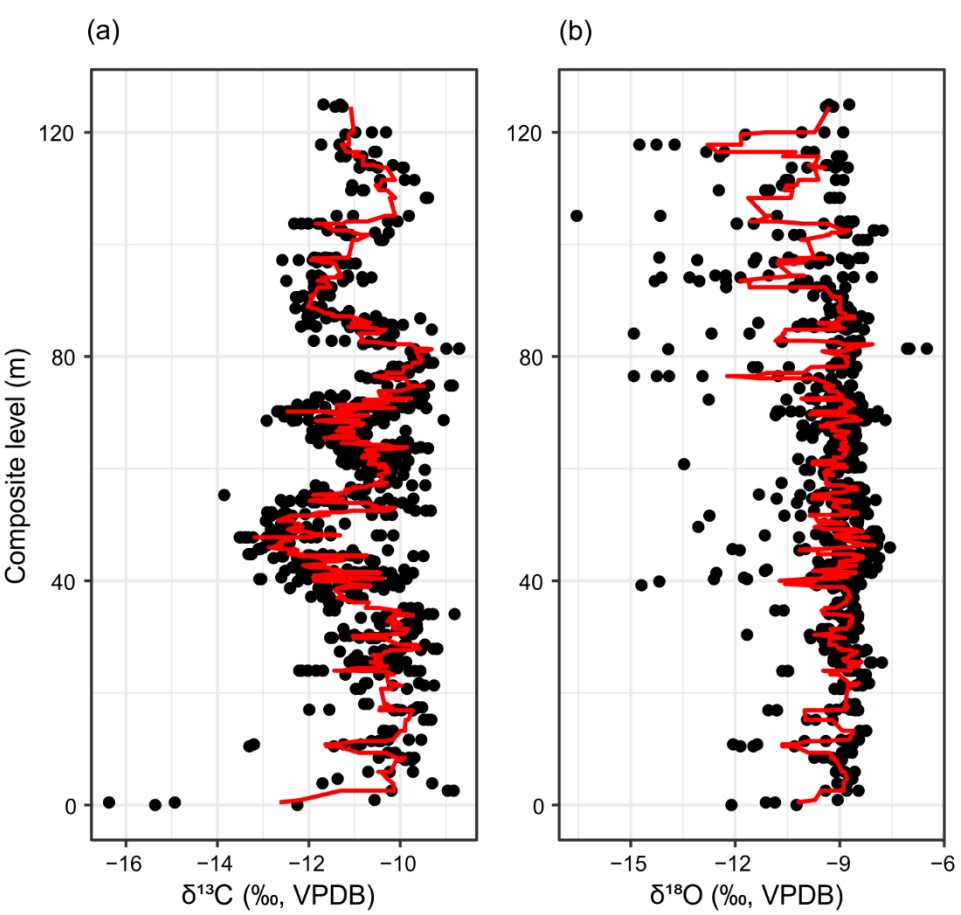

**Figure D1: Pedogenic carbonate $\delta^{13}C$ (a) and $\delta^{18}O$ (b) according to their composite stratigraphic level. Points represent individual sample measurements. Red line is a seven-point moving average through the values. This composite section includes the Basin Draw and D-1454 sections with the top of the Basin Draw section correlated to the lowest CIE (ETM2).**

**Data availability**

Raw and processed stable isotope data, as well as high-resolution maps and field photos and R scripts used for data processing and creating figures are available in Open Science Framework (OSF) and can be accessed here: https://osf.io/c4fgb

## Author contribution

W.C.C., R.S., and A.E.C. acquired the initial funding. S.J.W., W.C.C., R.S., and A.E.C., were involved in project conceptualization and fieldwork. S.J.W., R.S., and K.E.S. contributed to laboratory investigation and analyses. All authors contributed to writing and editing of the manuscript.

## Competing Interests

The authors declare that they have no conflict of interest.

## Acknowledgements

This project was supported by National Geographic Grant # 9969-16 to W.C.C., a University of New Hampshire Natural Resources and Earth Systems Science Graduate Student Research Grant to S.J.W., and funding from the University of Nebraska State Museum to R.S. We also thank L. Wingate and the UMSIL laboratory as well as B. Davidheiser-Kroll of the CUBES–SIL laboratory for stable isotope analyses, S. Kopf for development and implementation of IsoVerse R package used in the data reduction methods developed by CUBES–SIL, K. Rose, D. Todd, D. Hock, R. Gillham for field assistance, and M. Routhier for the differential GPS equipment and post-processing. We are also grateful to the Bureau of Land Management and Brent Breithaupt for providing Paleontological Resources Use Permits to A.E.C. and R.S. We thank C. Bataille, G. Bowen, and H. Abels for their helpful comments on an earlier draft of this work. Field work for this study was conducted on the ancestral and traditional lands of the Cheyenne, Crow, and Oceti Sakowin (Sioux), Eastern Shoshone, and Northern Arapahoe Indigenous peoples along with other Native tribes who call the Bighorn Basin and Rocky Mountain region home. We acknowledge and honor with gratitude the land and people who have stewarded it for generations.

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
