# Peer review of "Terrestrial carbon isotope stratigraphy and mammal turnover during post-PETM hyperthermals in the Bighorn Basin, Wyoming, USA"

_Climate of the Past, 2021_

## Author Response (AR1)

Below we have included a point-by-point response to the reviewer comments (Clement Bataille and Gabriel Bowen), as well as one detailed community comment (Hemmo Abels). One primary concern from all of the reviewers was to document the stratigraphy in more detail and show how individual sections and subsections have been correlated to one another, including the uncertainty in these correlations. We have addressed this concern by adding an additional appendix to the text (Appendix A). This appendix includes more detailed maps that show (1) sampling and bed-tracing trajectories for each section, (2) places were dGPS elevations were measured, (3) fossil localities correlated to our sections as well as other nearby fossil localities, (4) satellite imagery, and (5) elevation contours. In addition to these more detailed maps for each section, the appendix also includes figures that highlight the lithostratigraphy, sampling levels, fossil locality levels, dGPS levels, bed traces, and key marker beds for each section. The correlations between the six sections are also shown based on our bed traces and fossil locality correlations (Fig. A9). Lastly, the carbon isotope values are also shown for each section and represented with different symbols according to their subsection (Fig. A10). This provides a more detailed view of how isotope values vary over relatively short distances.

While creating this documentation, we divided the North Fork section from the previous version of the text into the new "North Fork" and an "Upper North Fork" sections. This was done to account for some additional uncertainty in the long bed trace between these two sections and ensure that each of our "sections" is made up only of shorter bed traces and correlations with minimal uncertainty. Our method of measuring the sections as a series of subsections tied to one another by bed traces is standard for fieldwork in sedimentary basins. We feel that with this new documentation, we've provided a level of detail that addresses reviewer concerns surrounding the individual sections.

Several comments also expressed concern over our composite stratigraphy and whether this could be averaging too many mismatched section correlations to be useful. To address this, we've removed the Basin Draw section from our composite section due to correlation uncertainties that are described in the text. Because of these uncertainties, we decided the evidence was not strong enough to correlate the CIE in the uppermost part of this section to ETM2. Thus, we leave the Basin Draw section uncorrelated and no longer include it in our baseline values for CIE magnitude calculations. This resulted in new CIE magnitudes that are more in line with those reported in Abels et al. (2016).

Our response to each of the reviewers' comments are included as bulleted points below in black text.

**Clement Bataille** (Referee Comment)

Widlansky et al. present much-needed chemostratigraphic information for the Fifteenmile Creek area covering some of the Eocene hyperthermals. There are two main objectives with this study: 1) compare the carbon isotope results with those of Abels et al. 2016 and assess the terrestrial amplification of the d13C signals and 2) provide a better chemostratigraphic context for some of the claims of Chew 2015 linking hyperthermal and faunal turnover. The authors

produce an impressive isotope chemostratigraphic record with hundreds of samples. The analytical issues encountered were a bit surprising but they are well-explained and justified. I believe this is a good paper with a lot of potential to better explore the link between hyperthermals, climate and biostratigraphy at the scale of an entire sedimentary basin. However, at this point, I don't think this study is fully reproducible mostly due to the lack of details relative to the stratigraphic work. I suggest major revisions to account for this limitation and make sure the findings can be compared to other locations across the Bighorn Basin

Major comments:

I recommend the authors to read for example "Lehman et al. Stratigraphy and depositional history of the Tornillo Group (Upper Cretaceous-Eocene) of West Texas". When I worked there, I used that document and could easily correlate stratigraphic sections with each other and reproduce the work of the authors thanks to maps, tie-points and field photos. This was super helpful to build upon their work, particularly the maps with the different sections and the stratigraphic correlation between sub-sections. You can also read the paper Bataille et al. 2016 Chemostratigraphic age model for the Tornillo Group: A possible link between fluvial stratigraphy and climate". I think your paper is more similar to Bataille et al. 2016 as it focuses mostly on chemostratigraphy and age model. I fully understand that this work is not a stratigraphic piece but I think that the study really needs to link with previous stratigraphic work in the area to strenghten the age model. So I am suggesting below some additional figures and material to help the readers use this new chemostratigraphic data.

It is ok to focus on chemostratigraphic correlations (e.g. Fig. $) but this needs to be done with a stratigraphic context. The authors should add a figure linking sub-sections with the selected tie-points and the method to correlate each section (see for example Fig. 6 in Lehman et al. or also Fig. 2 in Bataille et al. 2016 or supplement in Bataille et al. 2016). For example, we use the marker bed XX to correlate between sub-section XX and XX …etc… Or we used elevation records to trace this bed… So that the reader can understand solid tie points and more uncertain ones… If available some field photos of these marker beds would be really useful so that they can easily be identified in the field. Giving a table of tie-points used and their stratigraphic level would also be useful.

- We have broken the sections down further into individual subsections, along with their tie points in the new Appendix A (Figs. A1 – A6). We have also identified how each of the six sections are tied to one another (either by direct bed traces ourselves or using the Bown et al. (1994) composite meter level for fossil localities that are tied to the sections in this Appendix).
- We have also provided field photos of some representative outcrops in Fig. A7. These show examples of typical sampling strategies (photos a, b, and d), marker beds used for tracing (photos a, c, and d), and the relationship between sample sites and fossil localities (photo b).

- We have included a table of handheld GPS waypoints and differential GPS tie point coordinates and elevations in the Open Science Framework (OSF) data repository for this paper (https://osf.io/3z2xc/?view_only=1e9f419b168c43f5b0ae0b8e8f09584a).

The authors should add some zoomed maps in the supplement for each sub-section measured showing where they were measured (see for example Fig. 5 in Lehman et al.). This is really helpful to go back on the field. I understand there is GPS but this is easier to look at a path on a map in my opinion.

- We have added these zoomed in maps to the new Appendix A (Figs. A1 – A6). We felt it was best to show subsections from the same section together on one map so it would be easy to visualize how they were traced and measured together. There are now six maps to show each of the six sections, with their subsections labelled.

The author should add a full composite section of their subsections and compare it to Bown et al. 1994 with biostrat and magnetostatic data tie points.

- There is no composite isotope section in Bown et al. (1994) to compare our results, however, we have shown a full composite of our sections alongside magnetostratigraphy for the Fifteenmile Creek and Elk Creek Rim areas that was tied to the Bown et al. (1994) composite and faunal zone boundaries that are based on first and last appearance dates in Bown et al. (1994) (Figs. 4 and 6). These biostratigraphic and magnetostratigraphic constraints help to support our interpretation of CIE1 and CIE2 as EMT2 and H2, respectively.

In Fig. 5 it might be good to change the symbol by sub-sections to check if some of the noise is related to stratigraphic mismatch between sub-sections or at least to add this figure in the supplement. A broader discussion of stratigraphic mistmatch or uncertainties is also needed.

- We've added Fig. A10 that shows the isotope record for each section with different symbols for each subsection to help visualize how the isotope data correspond to the subsections. We've also included a figure in the supplemental R markdown file that includes the full composite section with different symbols for each section. This can be found in the OSF data repository.
- We've added discussion about the uncertainties in the composite section to the Discussion section 5.2

Once this is done it would be good to plot the chemostrat record using all the available age model information similar to Fig. 5 in Bataille et al. 2016… This is far from perfect but it helps the reader a lot in my opinion to link this record with broader record either from this basin or at the global scale.

- We've added the biostratigraphy and magnetostratigraphy to the top panel of Fig. 4, as these datasets were developed in the BCM framework. We've also added the

magnetostratigraphy to Fig. 6 so it can be compared to the McCullough Peaks and marine records and we've kept the Biohorizon B level in both terrestrial sections as the best biostratigraphic constraint for the levels.

I think these figures will also help the authors to improve a bit some of the discussion relating chemostratigraphy with faunal turnover and comparing chemostratigraphy with the Abels et al. 2016 section.

**Gabriel Bowen** (Referee Comment)

This manuscript reports new pedogenic carbonate isotope data from the Bighorn Basin in an attempt to link central/southern basin fossil localities to a global timescale and sequence of Eocene climate events. The substantial new dataset and goal of exploring within-basin variability in environmental proxy data and faunal change are strong aspects of this contribution. The manuscript also highlights challenging limitations to the correlation of local sections within their study area, which frankly leave some ambiguity with respect to the stratigraphic interpretations presented by the authors. I do think that the work has inherent value despite the uncertainties that remain, but given that the primary contributions of the manuscript derive from and rest on the correlations, I request that the authors try to further justify some of their reasoning and interpretations.

My biggest concern relates to the identification and interpretation of CIEs in the new records. In section 4.2 the authors identify a series of CIEs in their local study sections and implicitly propose and interpretation of those (that they are stratigraphically coherent and can be correlated between sections, and are therefore likely to be reflective of large-scale or global forcing). I think that both parts of this presentation need stronger support and justification.

- We've restructured the Results section to (1) present the stable isotope data (Section 4.1), (2) describe the CIEs found in each section individually and the field correlations that were used to support our conclusion that there are 3 CIE intervals (Section 4.2), and (3) show how the data look when they are transformed into either the BCM or elevation framework (4.3). We feel that this gives a better representation of the methods and reasoning we used to correlate among these sections. The identification of 3 CIEs is supported by field observations and knowledge of the local stratigraphy, which is presented in section 4.2 and Appendix A. Our reasoning and methods were not always clear in the previous version, which may have led the reader to think the correlations were based purely on BCM levels and/or elevation.
- The conclusion that these CIEs are related to large-scale or global forcing is based on the co-occurrence of the CIEs and the magnetic reversal/Biohorizon B constraints in Fifteenmile Creek as well as other locations in the basin (based on magnetostratigraphy and biostratigraphy) and marine records (based on magnetostratigraphy).

First, the authors need to more clearly describe what criteria they identify excursions in their records. Like many continental records, the ones presented here are noisy, and their features

are not always obvious. Some of the excursions are pretty obvious (e.g., the one in the "Basal" local section). Others less so (e.g., why is the feature at ~440 BCM in the Basin Draw section a CIE and the one at ~400 BCM not?). I understand that it may be challenging or impossible to offer a fully objective and quantitative set of metrics used to guide these interpretations, but I'd like to see the authors try to get as close as possible to this, and then elaborate and justify other information used to (more subjectively) guide their interpretation. Since the identification of the CIEs is the major contribution of the paper, this deserves more attention.

- We've added more explicit text that describes our criteria for identifying a CIE. Lines 270-273 now read, "Carbon isotope excursions were identified as places where site mean $\delta^{13}$C values were less than -12 ‰ and the values show a clear decrease and return to background levels (between about -9 ‰ and -11 ‰, mean background $\delta^{13}$C = -10.3 ‰). Places where single samples or site means were less than -12 ‰ and were surrounded by background values above and below or places where the return to background values was not captured were not considered CIEs."
- In places where we used more subjective criteria for identifying or excluding something as a CIE, we've made that clear. For example, lines 279-285: "The base of the North Fork section includes sites with relatively low $\delta^{13}$C values (minimum site $\delta^{13}$C = -13.3 ‰) that return to background levels around the 20 m level. Based on field correlations and close geographic proximity with the Basal/D-1350 section (Appendix A), we interpret this negative shift to be equivalent to the excursion at the top of the Basal/D-1350 section. This interpretation is also supported by composite meter levels from Bown et al. (1994), which indicate that stratigraphic levels at the top of the Basal/D-1350 section should be equivalent to levels in the bottom of the North Fork section. Thus, we consider the negative shift in values at the bottom of the North Fork section to represent a CIE, even though the onset is not captured here."
- The middle part of the Basin Draw section does not show a clear excursion and recovery, and instead appears to be part of a generally low $\delta^{13}$C zone with no clear pattern, so we do not consider it a CIE.

Second, the correlation of different CIEs between sections is implied in section 4.2, and then revisited in the discussion sections. The initial presentation implicitly accepts that the BCM levels are an accurate basis for correlation between sections (the identification of the CIEs and association between them is introduced in terms of their being associated with certain BCM intervals). This is then revisited and questioned, and a second basis for correlation (GPS elevations) introduced and discussed. Arguments are made for preferring the GPS data in some cases (e.g., correlation from the N to S sides of the wash). Given that bed tracing is difficult in the area, and that GPS does not account for structural and depositional surface features, we're left with a pretty inconclusive case. I'm left wondering why two other potentially-useful sources of information, namely patterns in the CIE records and fossil evidence, aren't used in the correlation exercise. For example, both the up-section trend and the relative amplitude of the excursions (IMO) strongly support the correlation of the two CIEs at Kraus Flats with the lower two features in the North Fork section (and would imply slight deviations from both the BCM and GPS-based correlations).

- This is also addressed in an earlier comment. The new section 4.2 presents our evidence for correlating the CIEs that are based on field observations, biostratigraphy, and chemostratigraphy, before presenting the BCM and dGPS results (section 4.3). This includes correlation between the two CIEs in Kraus Flats and North Fork sections.
- Mammalian biostratigraphy during this time interval is not resolved finely enough to be useful for correlation at this scale of resolution and relatively few individual localities are directly correlated to our sections so any FAD or LAD based on this group of localities alone would be highly uncertain.

Conversely, I continue to struggle with the proposed correlation between Basin Draw and Basal sections given the strong isotopic 'structure' exhibited in the lower part of Basin Draw (and not at Basal). The authors have discussed local effects on soil carbonate d13C at length, and it's possible that's what we're seeing here, but I'm not so sure. In addition, the authors are drawn to work in this area because of the extensive history of fossil collecting, and there are several well-described faunal turnover events documented in their study sections. The text implies that there may be some noise or ambiguity in the pattern of turnover that might in part reflect incorrect correlation between sections (that could be resolved here)…why isn't this information tapped and discussed/leveraged as part of building the correlation model here? Finally, what about lithostratigraphy? Are there any coherent patterns that might support the correlation model?

- Due to the uncertainty of the correlation between Basin Draw and the other sections, as well as the complicated cut-and-fill deposition near the top of the Basin Draw section and noisy isotope record from the same area, we no longer feel the evidence is strong enough to correlate the Basin Draw CIE to excursions other sections.
- Using the new stratigraphic framework presented here to better resolve the faunal turnover across Biohorizon B is an important goal of our research program but it is outside of the scope of this study. In this study, we aim to develop a reliable chemostratigraphic framework for this research area and relate it to previous studies of faunal turnover. Direct correlation of many additional fossil localities to this new chemostratigraphic framework will be necessary to completely redo the faunal analysis and further interogate the relationship between biotic change and these CIEs
- There are no coherent lithostratigraphic patterns that can be traced between Basin Draw and the other sections.

In summary, I think that this will be a nice contribution to our understanding of the stratigraphy of the BHB and advance our ability to link environmental and biotic events in the basin to global changes. I think some revision focused on shoring up interpretation of the new carbon isotope stratigraphy is need to accomplish that. As a final thought, I'm not sure that the discussion of the local controls on soil carbonate C isotope values adds much (it is important background, but doesn't really support any clear or important conclusions from this work)…this could be cut substantially, IMHO, and some of the space allocated toward shoring up the stratigraphic interpretations.

- We've removed this entire section, particularly because our interpretations are now less focused on explaining the differences between Fifteenmile Creek and other records. We've included relevant information from this section when needed in section 5.3 where we compare Fifteenmile Creek to McCullough Peaks.

Minor points:

Abstract: here 15 Mile Creek is described as being in the 'central Bighorn Basin', whereas throughout the introduction it is stated to be in the 'southern Bighorn Basin'

- We've changed the wording to be "south-central" in the abstract and text. This wording will help it stay consistent with other work from the area as well (e.g., Chew, 2015).

Section 3.2: Were the nodules collected from/tied to individual paleosol B horizons? Was any attempt made to constrain depth below the paleo-soil surface?

- We've updated the text in lines 188-191 to read, "Sample sites were chosen to ensure at least a 1-meter sampling resolution, where possible, and levels near a potential CIE were sampled at a higher resolution during subsequent field seasons. Samples were collected from > 30 cm below the upper surface of B horizons in variegated paleosols. At this depth, soil $CO_2$ is primarily plant-respired and will track atmospheric $CO_2$ $\delta^{13}C$ (Cerling, 1984)."

L233: Cite these packages if you're going to mention them here

- We've added citations to the Tidyverse paper (Wickham et al., 2019), IsoReader (Kopf et al., 2021), and R (R core team (2019).

L331-333: This makes it unclear why you prefer to associate the Basin Draw CIE with the lowest of the three events (as shown in Fig 4).

- We no longer correlate the CIE at Basin Draw to any other sections, as discussed above. We still present our isotope results from Basin Draw to help inform future sampling and additional research in this area.

L409-421: These arguments are reasonable but not particularly strong…for example one could argue that the d13C values around the ~30m level in the Basin Draw local section represent the H1 CIE. What does the fossil evidence say? Is there anything there that provides evidence for the alignment of this section wrt the faunal events?

- The fossil assemblages from this section are ambiguous in terms of helping with correlation. As stated above, we feel that the ~30 m level of the Basin Draw section did not show a clear excursion and recovery, and instead appeared to be part of a generally low $\delta^{13}C$ zone with no clear pattern, so we did not consider it a CIE. Based on the BCM

levels for the D-1454/D-1460 fossil localities (~35 m in the Basin Draw section), we would expect those localities to be within ETM2, although this is ~5 m higher in the section than where we see more negative $\delta^{13}C$ values.

Paragraph starting on L422: Is there value in doing this work without recollecting the fossils with higher stratigraphic resolution? High-resolution isotope data combined with ambiguous fossil locality extents is still likely to yield confusion.

- While a high-resolution fossil record paired with a high-resolution isotope record would be ideal, it is well beyond the scope of this project. However, we still see this study as an opportunity to provide the first high-resolution isotope record for the area, and to highlight places where the fossil collecting and documentation could be nailed down better to later produce a new faunal record that is more closely tied to the isotope stratigraphy.

L476-479: This explanation isn't quite right…atmospheric CO2 isn't 'penetrating deeper into the soil', it's always there at all depths. It's just that it constitutes a larger fraction of the total CO2 in this situation. (Correct this in the subsequent paragraph, too.)

- We've taken this section out, but this is helpful for future reference.

**Hemmo Abels** (Community Review)

Herewith my review of Widlanski et al. submitted to Climate of the Past.

The paper brings an important subject interesting for a relatively wide audience of paleontologist and paleoclimate workers, both terrestrial and marine. The early Eocene hyperthermals are well studied while little data are present concerning their impact on the continents both geochemically, as environmentally and faunally. The paper is a continuation of earlier work published in Climate of the Past (Chew 2015) where an attempt was made to correlate faunal records of the central Bighorn basin, Wyoming, to a carbon isotope record from more northern parts of the same basin. The faunal analysis of that previous paper is now supplemented by a series of carbon isotope sections much closer or directly at the mammal sites in the central parts of the basin and with that potentially making the correlations between fauna and isotope changes more straightforward. The authors face the problem that, in the central parts of the basin, outcrops and so carbon isotope samples and mammal sites are scattered over large areas due to low topography. A nearly 30-year old composite meter level system is used since then to anchor different data sets to the meter-level system over this entire area, while structural dips are reported to occassionally change within the area without much notice. The composite level stratigraphy (BCM named here) has thus a relative large uncertainty and mammal sites and carbon isotope records may erroneously overlap or be separated in stratigraphy while in reality that is not the case. Therefore, the current authors additionally use absolute dGPS levels within their sections to also tie sections together. Subsequently, a composite isotope stratigraphy is constructed where the mammal analysis of

Chew 2015 is placed along and mammal and isotope changes are discussed and placed in a more global perspective.

Correct stratigraphy is thus the key to produce solid results in this paper as in many other papers. The difficult outcrops make that the stratigraphic approach and data should be produced and communicated with even more care. How do isotope and mammal finds stack together into stratigraphy? The authors must have spend considerable time to produce the current work, however, the written work under review here did not take away my concerns about the mammal and isotope composite stratigraphy produced. I think this is partly because the workflow and data are not sufficiently backed-up with maps and data in the paper or supplement.

- The new Appendix A contains additional detailed maps and figures showing the subsections, bed traces, sampling levels, correlated fossil localities, and dGPS tie points. Data tables with handheld GPS waypoints and differential GPS coordinates with elevations have also been added to the supplement. Additionally, we've added more detail about the biostratigraphy to Figs. 3 and 4 to help place the mammal fauna in the chemostratigraphic framework.

There are large stratigraphic thickness changes between steps within the workflow of up to 30% for individual sections. How is that possible? One would expect that dGPS data or Jacob-Staff data should be able to work at lower uncertainty? Were structural dips of the layering not determined sufficiently?

- We've added a section to the results that details these stratigraphic thickness differences (Section 4.3: Stratigraphy). While it is not ideal to see so much variation between methods, we feel this is the reality of working in an area with low-lying and sometimes semi-isolated outcrops. Within a section, Jacob Staff measurements provide the least uncertainty and give a confident measurement of thickness. Correlation into the Bown et al. (1994) framework is necessary to relate our findings to earlier work on faunal change (e.g., Chew, 2015) and also provides one means of correlating among our sections. While measuring our sections, we noticed places where our measured thickness between fossil localities did not match those of Bown et al. (1994), so it is not surprising that transforming the data into this framework would produce some minor differences. Differential GPS elevations were used as a second means of correlating among sections and to verify the correlation using the Bown et al. (1994) stratigraphy. These two methods have their own associated uncertainty, as discussed in the text, but they each independently confirm the superposition of the observed CIEs. When creating the composite section (Figure 6) we chose to correlate the CIEs by lining up the minimum excursion values and using the thicknesses that were measured by us using a Jacob's staff as these are the most reliable and have not been adjusted to fit either the BCM or elevation frameworks. We've made this more clear in lines 404-412 that now read, "To account for the geographic variability in bed thickness described above, the five stratigraphic sections that are north of Fifteenmile Creek (Basal/D-1350, Kraus Flats,

North Fork, Upper North Fork, and Red Butte), as well as the shorter sections through fossil localities D-1207 and D-1532, were compiled into a "Fifteenmile Creek Composite Section" (Fig. 6). To create the composite section, the peak negative $\delta^{13}$C values from each CIE were aligned and the relative spacing between CIEs was scaled to the average of the spacing from the sections where we measured the CIEs together. All stratigraphic thicknesses for the composite section are based on the thicknesses we measured with a Jacob's staff in the field, rather than using BCM levels or elevations. The Jacob's staff measurements offer the least uncertainty for individual sections and isotope values are the best means for correlating between sections for purposes of creating a composite section. These correlations are independently supported by both the BCM and elevation frameworks."

- When measuring sections with a Jacob's staff we usually assumed a dip of zero degrees. In local areas where strike and dip could be measured, we found that dip was very close to zero. When dip is close to zero, it is extremely difficult to accurately calculate the direction of strike, especially in semi-isolated badland outcrops like those in the field area. Thus, we do not have measurements of structural dip that can be placed on the maps. Most variation in dip in the Fifteenmile Creek area is minor and localized and does not reflect broader structural patterns. If local differences in dip along the section transect were detected (e.g., by shooting a horizontal line between exposures of a marker bed along the transect), adjustments to dip were made on the Jacob Staff's clinometer. It should be noted that in most cases, section thickness was measured on slopes of 30 degrees or more, where beds were well exposed, and small differences in dip would have had only a small effect on overall section thickness. These small, local subsections were then tied together with bed traces, sometimes over long distances, and these long bed traces are likely where most of the uncertainty lies.

The isotope records are at places of relatively high resolution, although there is no explanation of sampling strategy. Where all pedogenic nodules sampled?

- This is addressed in a later comment.

The carbon isotope records show some very good CIEs, but also some interval difficult to interpret. The resulting composite isotope record however is far from easy to interpret. Due to the stacking of series that do not have very similar carbon isotope results, the composite stratigraphy is blurry. Relating CIE1 to ETM2 and CIE2 to H2 seems logic with the additional stratigraphic constraints, but I wonder whether correlation to I1-I2 would be possible? CIE sizes have so-far been showing to be very stable, such as even similar PETM CIE body d13Cped values in the northern basin as in the southern basin. The paper has long discussion on what could have caused different CIE sizes within the basin, but while I am still doubting the stratigraphy, interpretations and composite stratigraphy, it is difficult to believe the remainder of the paper.

- We have removed Basin Draw and the small D-1454 section from our composite figure (Fig. 6) due to the uncertainty of correlating across Fifteenmile Creek that is discussed in the text. This takes away some of the noise near the base of the composite section and

the lowest CIE. We've also added magnetostratigraphy from the Fifteenmile Creek/Elk Creek Rim area (Tauxe et al., 1994; Clyde et al., 2007) to the composite section that further strengthens our interpretation.

- The position of the lowest excursion relative to Biohorizon B precludes correlation of the lowest two excursions to I1 and I2. Biohorizon B occurs over the 370 – 394 m level in the existing Bown et al. (1994) stratigraphy. This level is included at the base of our Basal/D-1350 section. There is no evidence for an unconformity between that level and the CIE in this section and the high resolution of sample levels (Fig. A2) makes it unlikely that the ETM2 and H2 CIEs were not captured. The magnetostratigraphy also supports this correlation as CIE1 and CIE2 fall within an interval of ambiguous polarity around the C24r – C24n.3n magnetic reversal that is consistent with the placement of ETM2 and H2 in other records.

The seeming lack of documentation of stratigraphic data within the paper and appendix can be solved. Mammal sites, carbon isotope sections, and magnetostratigraphic sections should be plotted on a map(s) that is(are) detailed enough to see how different site (may) relate. Figure 1 is small and the background of it vague. In the supplement or paper, more detailed maps could help to show such relations. In these map figures, it is necessary to show topographic lines, the key mammal sites, the isotope sections and the structural dip of layering. This allows to understand how these result in stratigraphy.

- We have added more detailed maps of each section to the new Appendix A and also updated the map in Fig. 1 to include topographic lines and landmarks like Fifteenmile Creek and Fifteenmile Creek Rd. Due to the reasons discussed in an earlier comment, we do not have structural dip measurements to add to the map. The detailed section maps are shown along with the stratigraphy for each section to help place the map into a stratigraphic context and subsections, bed traces, fossil localities and their traces, and dGPS tie points are shown on the maps as well.

Besides, photographs of the sections sampled should be added such to see the outcrops and sampling trajectories. Are these far away even within single sections? It seems so, if I see the North Fork section on Figure 1 and try to find it using Google Earth, I only see very low hills with 5-10 meters of stratigraphy. What is the uncertainty in thickness of single sections when measured in the field by Jacob Staff?

- We have added photographs of representative sampling areas to Appendix A (Fig. A7). These show examples of typical sampling strategies (photos a, b, and d), marker beds used for tracing (photos a, c, and d), and the relationship between sample sites and fossil localities (photo b).
- It is correct that many of our sections are comprised of small (5 – 10 m) subsections. Beds are well exposed in individual subsections, so Jacob's staff measurements have a low uncertainty. Since Jacob's staff measurements were taken on the steepest exposures available in local subsections, measured thickness error is fairly small, and probably less than 10% in most cases. However, bed thickness can vary by up to a few

meters in this area. Bed tracing were usually done using the base of a bed as a stratigraphic datum, but beds sometimes grade laterally into different colors representing different degrees of soil development on different parts of the floodplain, which can cause the level of the bed's base to vary geographically. Therefore, longer bed traces between sections can have an uncertainty up to ± 10 m. Places where these traces may have large uncertainties are shown in Figs. A1 – A6.

There misses a table with the dGPS data that are used and including GPS locations of the tops and bases of the (sub-)sections.

- These data were included in the OSF data repository for this paper, however they were confusing to access as they were formatted to be used in the R code for plotting. We've added tables with dGPS and handheld GPS data that can be accessed more easily. ([https://osf.io/3z2xc/?view_only=1e9f419b168c43f5b0ae0b8e8f09584a](https://osf.io/3z2xc/?view_only=1e9f419b168c43f5b0ae0b8e8f09584a)).

The Results section misses a detailed description of stratigraphy and stratigraphic results. The dGPS stratigraphies differ quite a bit from the BCM levels supplemented by Jacob Staff measurements. In Figure 4, both stratigraphies are provided, there is a description of the isotope stratigraphies of both methods to come to stratigraphy, but there lacks a description of the stratigraphic impact itself between the two methods. Basin Draw is shortened by 33% it seems in the dGPS method, Kraus Flats is shortened by 20%, while North Fork is expaned by 33%. One would expect that Jacob-Staff data are not that inaccurate? Why would this happen?

- We've added a new section (4.3) to present the stratigraphic results and this section includes a detailed description of the thickness differences between the three methods (Jacob's staff, BCM, and elevation). The thickness differences within an individual section are likely not related to inaccuracy in the Jacob's staff measurements. Instead, these are more likely due to uncertainty in the BCM and elevation frameworks that are discussed in the text.

Another argument could be that the approximate levels of the dGPS are indeed reliable, so still lining up the supposed ETM2 level, but for thicknesses the Jacob Staff data are quite good?

- Our dGPS method still uses the relative thicknesses measured using a Jacob's staff to determine the elevation for sites that are between dGPS tie points. This concept is explained in the new Fig. A8. For sections with more dGPS tie points (e.g., Kraus Flats) the total thickness in the elevation framework is going to be influenced more by the elevation measurements. For sections like Basal/D-1350 with fewer dGPS tie points, the tie points are used to correlate the excursion and marker beds within the D-1350 locality, and the total thickness is determined from the Jacob's staff measurements. We recognize that this creates some differences in the way individual sections are shown in the elevation framework and this is why we chose to use Jacob staff measurements and isotope correlations to create our composite framework. The dGPS and BCM framework

simply allow us to confirm which excursions should be correlated to each other and to tie our results in to the surrounding fossil localities.

- This is addressed in an earlier comment.

In relation, there misses an impact on mammal site stratigraphy of the two stratigraphic compilations made. Part of these results are presented in the discussion, but these should really be in the Results section. Missing is a table with mammal site content in the new stratigraphic order listing the most important mammal species through stratigraphy, it must be different with these new approaches from Chew, 2015? Grey bars indicate BioB, B1 and B2, but the mammal sites where this is based upon should provided.

- We've moved the results relating locality levels to the CIEs to the Results Section 4.4 (lines 344-356). We feel that a longer discussion about the implications of these findings for the Chew (2015) study are more appropriate in the Discussion section and are found in Section 5.4.
- We think that re-doing the faunal analyses of Chew (2015) is a worthy topic separate from the main focus of this paper, which is to identify CIEs in the Fifteenmile Creek area. Such a study will require the correlation of many more fossil localities into a broader chemostratigraphy in order to have enough fossils incorporated to be meaningful. We have, however, added more information about the fauna from the fossil localities that were correlated to our sections in Fig. 3 and added the faunal zones to the top panel of Fig. 4.

In addition, the authors do not include the notion of precession forcing of stratigraphy on the floodplain stratigraphy of the Bighorn Basin. Abdul Aziz et al. 2008 suggest precession forcing in what seems one of the currently studied sections, Red Butte. 20 kyr would correspond roughly to 8 meters of section according to the work of Abdul Aziz. For more northern sites, such numbers have been documented since. Also, Abels et al. 2016 discuss a ca 35m cycle in the carbon isotope record studied that would be in line with the precession forcing of the smaller scale cycles. Both these numbers give a fairly good control on sedimentation rates and from that point of view the isotope records could be analysed and interpreted. This is not used nor discussed. Line 353-355 are a little too easily stating that there are "overall differences in sediment thickness across the basin". Yes, there are clearly differences, and also hiatuses particularly at the basin margins, but sedimentation rates in the basin centres have been shown to be relatively constant. In the centres, sed rates depend on subsidence rates that controlled net accommodation space generation. One would not expect rapid changes in sedimentation rates at above $10^4$ yr and below $10^6$ yr time scales as are suggested by the correlation of CIE3 to I1. There is discussion on this in 5.2 and 5.3, but it is quite unstructured now also discussing the same things in different sections (line 360-366 in 5.3 and similar discussion with different arguments in 5.2). As a minimum, the authors should refer to the work on astronomical forcing of these series and why they think not to use these arguments in building

stratigraphic framework for their series and for the interpretation of their carbon isotope records. The authors claim lower sedimentation rates by observing lower spacing between CIE1 and CIE2 in their series. If these are correlated to CIEs in marine records, it also imports 100-kyr eccentricity age control on the Fifteenmile Creek series that could be used to interpret the remainder of the series.

- The Polecat Bench and Red Butte sections of Aziz et al. (2008) each have relatively continuous exposures where incised channel sandstones can be avoided that make them ideal for identification of cycles related to astronomical forcing. However, this is not the case for our sections. Conducting similar analyses here would introduce so much uncertainty as to make the results uninterpretable.
- We have added discussion about the astronomical forcing constraints on sedimentation rates to Section 5.2 (lines 437-446). The Red Butte section of Aziz et al. (2008) is located ~3 km northwest of our Red Butte section, which can offer constraints on the sedimentation rates for the upper portion of our section. The spacing between ETM2 and H2 and eccentricity cycles associated with these CIEs give a ~4.5 kyr/m sedimentation rate for the lower part of our section. Assuming that we have a fairly continuous record and using these two estimates as constraints on the sedimentation rates in our section, it is unlikely that our CIE3 would correlate to I1.

The second half of section 5.3 does not include the possibility that there may be remaining uncertainties in the study. There are at least some serious stratigraphic doubts to be placed along the composite stratigraphy with the dGPS stratigraphy so largely different from the Jacob's Staffed thicknesses. If the thicknesses of individual section can be different up to 30% between methods because of uncertainties in structural dip changing through the area and because of very low topography of outcrops, correlations between sections must contain serious uncertainties.

- We've added a new figure that shows the correlations between sections based on our own bed traces and fossil locality correlation (Fig. A9). These "field-based" correlations are supported by the BCM and elevation frameworks and confirm our identification of the three CIEs.
- The correlations between sections to create our composite are based on lining up minimum excursion values. The stratigraphic thicknesses in the composite are based on Jacob's staff measurements. Although there may be some uncertainty in the Jacob's staff measurements, these will be consistent across the sections and there is no mixing of BCM and elevation stratigraphy into the composite that would add additional uncertainty. The thicknesses of individual sections will be equal in the composite and local sections measured with a Jacob's staff. The exceptions to this are the Kraus Flats and North Fork sections that record both ETM2 and H2. 21.3 m separate the two peak excursions in Kraus Flats and 23.6 m separate the excursions in North Fork. An average of 22.5 m was used as the spacing between these CIEs in the composite and the levels in each both sections were adjusted to maintain this spacing. This average was very close to the thickness in each of the two sections, so it does not significantly alter either of

these records in the composite and this can be considered a reliable estimate of the thickness between ETM2 and H2.

- There is some uncertainty in the placement of CIE3 in the composite because there is no place where CIE2 and CIE3 were measured together in the same section. The North Fork and Upper North Fork sections are tied together by a bed trace and the thickness between CIE2 and CIE3 in the composite section is based on this measurement to remain consistent with the way we've constructed the composite for the rest of the record. The elevation and BCM frameworks each suggest an additional 20 meters between CIE2 and CIE3 compared to the thickness we measured. We added text that discusses this uncertainty in the placement of CIE3 to lines 448-456.

The composite isotope record is far from clean likely because of these uncertainties. Thicknesses of intervals and sections are different also in the composite. How can the authors be certain about the composite? Should the composite actually be made or is it better to discuss the results from the individual sections as those are uncertain enough in themselves?

- We feel that having a composite section facilitates comparison between our record and others, such as McCullough Peaks and the marine record. Some of the noise in our composite section was removed by taking out the Basin Draw and D-1454 sections that are located south of Fifteenmile Creek and had some additional uncertainty in their correlation.
- We feel that our individual sections have no more uncertainty than other studies that use Jacob's staff measurements in similar terrain. Lining up the isotope records to create a composite is also routinely done, so our composite section does not include any of the uncertainties that are associated with the BCM or elevation frameworks.

What could improve the carbon isotope data analysis is nailing down better the baseline carbon isotope values. This is around -10 per mille it seems. If a vertical line is placed at the -10 permille d13C value in all plots, it much better allows to identify excursions, also when records are relatively short.

- We've added dashed lines at -10 ‰ in Figs. 3, 6 and A10 to help visualize the average baseline values alongside the isotope results.

The BioB, B1 and B2 'events' are used without much discussion on their reliability. There should be minimally some discussion on these data and interpretations, such as why sample size and different types of diversity go hand-in-hand in the records of Chew 2015. Plotting the average sample size plot on top of the diversity plots in the Figure 4 of Chew, 2015, lets them merge, which despite the statistics places some serious doubts along the B1 and B2 events. And, does the new stratigraphy not impact those previous results warranting a new analysis?

- Re-doing the faunal analyses of Chew (2015) is outside the scope of this project, although we do recognize it as a promising area for future work. Our primary goal was to identify the CIEs in the Fifteenmile Creek area and a secondary goal was to test the

hypothesis proposed by Chew (2015) – that the interval associated with the B-1 and B-2 events could be associated with the ETM2 and H2 hyperthermals. Our results generally support this with some uncertainty that is described in the text.

- Of all of the standardized paleoecological parameters calculated by Chew (2015) from extensively size-standardized sample bins (including first and last appearance rates, evenness, dominance, alpha richness, beta richness, and relative abundance/body size), only beta richness was strongly significantly correlated with the original non-standardized sampling distribution (alpha richness was weakly correlated). The balance of evidence, including previous work (Chew and Oheim, 2013) in which a significant relationship between beta richness and non-standardized sample size was noted even in data that were extensively standardized for both sample-size and geographical area, suggested that sample-size bias was adequately controlled, permitting biological interpretation of the paleoecological parameters.
- We have added text that describes the potential influence of sampling bias in Chew (2015) to lines 136-138: "Species richness parameters used by Chew (2015) were significantly correlated to sample sizes, potentially indicating some sample size bias. Chew and Oheim (2013), however, found similar increases in beta richness after correcting for sample variation in the same study area, suggesting that these patterns are independent."

the title is too general now and should be refined to cover the content of the paper. The paper is not improving carbon isotope stratigraphy of post-PETM time intervals. A reference to 'terrestrial' or 'continental' and the geographic location is needed.

- We've updated the title to reference the geographic location (Bighorn Basin, Wyoming, USA) and added that these are terrestrial carbon isotope records.

in Figure 1. Would there be the possibility to both show topographic lines and structural dip of bedding? It would give some more notion of the stratigraphic transects / sections measured

- This is addressed in an earlier comment.

in Figure 1. It would be good to add some trails / tracks / roads to provide the reader the opportunity to better orient

- Fifteenmile Creek Rd. was added to the map in Figure 1 and the map for the Basal/D-1350 section (Fig. A2) as this is the most reliable road for navigating through the field area. We've also added Fifteenmile Creek for reference in Fig. 1.

in Figure 1. Legend is missing the light brown shading, from the caption that seems to be 'unshaded' Quaternary?

- We've changed the basemap for Fig. 1 and the detailed maps in Appendix A to only use Google Earth Satellite imagery as we feel this is the best way to visualize the exposures that were sampled. The geological map of Green and Drouillard (1994) that was

previously used was done at a different spatial resolution, and at this scale it misrepresents some fossil localities as being in Quaternary alluvium, when they are in fact in the Eocene Willwood Formation.

section 3.1 is unnecessary, items can be discussed when applicable in the results interpretation or discussion sections

- We removed this section and renamed Section 3 to be "Methods" instead of "Materials and methods".

Line 196. It is not clear what was the strategy to come to 'sampling sites'. Where all nodules encountered sampled, or was a certain stratigraphic resolution chosen?

- We have updated the text in lines 188-191 to say, "Sample sites were chosen to ensure at least a 1-meter sampling resolution, where possible, and levels near a potential CIE were sampled at a higher resolution during subsequent field seasons. Samples were collected from > 30 cm below the upper surface of B horizons in variegated paleosols. At this depth, soil $CO_2$ is primarily plant-respired and will track atmospheric $CO_2$ $\delta^{13}C$ (Cerling, 1984)."

Line 212 refers to pedogenic carbonate sampling again, right? Should it be in 3.2 then?

- We removed section 3.2 Fossil localities and added this text to the end of Section 3.1 Field methods (lines 205-209). We also changed the wording to make it clear that this "sampling" is referring to carbonate nodules (line 207).

Line 276. I would not regard the whole range between -8 and -12 per mille d13C as background. There is clearly noise on these records, but not 4 per mille.

- We used $\delta^{13}C$ values reaching -12 ‰ or lower as one criterion for identifying a CIE, however we did not use the entire range of -8 to -12 ‰ as background values when we detrended the data to come up with CIE magnitudes. The background values for this calculation fell between ~-9 and -11 ‰. There is also a figure included in the R markdown in the OSF data repository that shows these background values relative to the excursions. We've updated the text in lines 270-273 to reflect this. The new text reads, "Carbon isotope excursions were identified as places where site mean $\delta^{13}C$ values were less than -12 ‰ and the values show a clear decrease and return to background levels (between about -9 ‰ and -11 ‰, mean background $\delta^{13}C$ = -10.3 ‰), Places where single samples or site means were less than -12 ‰ and were surrounded by background values above and below, or places where the return to background values was not captured were not considered CIEs."

Line 275-280. Plotting against Trimble positions suggests there is no structural dip in the area?

- We recognize that this assumption may not give the best representation of true stratigraphic thicknesses across the field area. However, we feel it is sufficient to confirm our field correlations and it provides a framework for correlating exposures in the area that is independent of the biostratigraphy. As stated in an earlier comment, we do not use elevations to come up with our composite stratigraphy.

Figure 3. 'colors represent outcrop color' is not a very clear statement, I suppose you mean after removing weathered surface material? Or you mean color of the weathered surface?

- We've taken the lithostratigraphy out of Fig. 3 and moved it to the Figs. A1 – A6 where it can be shown in more detail for each section. In these new figures, the caption reads, "Colors represent the surface color of weathered rocks." We recorded surface colors as these are the most useful for bed tracing.

Figure 1 and 3. How does the Red Butte section correlate to the Red butte section of Abdul Aziz et al. 2008 in Geology? As the same name is used here, and it seems to be in the same area, it should be indicated how these relate, or another name should be used for the section. There are workers around who can help identify the location of that previous Red Butte section.

- Red Butte is a prominent feature in the field area. The Red Butte section of Aziz et al. (2008) and our Red Butte section are both located on the southern side of Red Butte, and ours is ~3 km south of theirs. We've indicated this in the text line 438.

Figure 3: why is the level at 45m in North Fork not identified as a CIE? And 35m and above in Red Butte? It is clearly not baseline of -10 per mille

- The 45 m level in North Fork (now part of the Upper North Fork section) was identified as the highest CIE in our section (CIE3). Because the peak values for this CIE were clustered so close together, the blue rectangle that represents the range of values included in the CIE mean is covered by the data points. We've added text to the caption for Fig. 3 to highlight this: "Note that the CIE at ~45 m in the North Fork/Upper North Fork section includes a small range of ~-12 ‰ values."
- We do mention in lines 288 and 414 that the top ~10 meters of Red Butte appear to be capturing the onset of another excursion. We do not calculate a magnitude for this excursion, however, because we did not capture the full onset and recovery. These samples were not used to calculate the baseline averages or to detrend the data.

Figure 3: in Basin Draw D1459 is given as 29 meters from D1460, but in the panel c of Basin Draw it is no more than 15m without visible uncertainty, how is that possible? The same occurs with D1822 and D1204top, where a separation is given of 18m while in the panel c it is no more than 4 meters

- This could happen in places where we did not know the precise bed that Bown et al. used to construct their composite section. Some fossil localities include several meters

of section and correlating a different bed into our section would result in different thicknesses between localities. In places where we were able to identify a range of collecting levels, we've shown this uncertainty in Fig. 3, panel b, but this was not always possible. Bown et al. also would have followed different trajectories when measuring their sections, so variations in bed thickness could also play a role.

Figure 3: it would be good to label the levels in panels c with the 1,2,3,4 numbers

- We've made this change and also indicated when taxa that constrain the biostratigraphy have been found at these localities.

Figure 3: Basal/D1350 has only few dGPS points while it is 50 m thick, why are there so few points and is the whole section not covered by calibration points?

- The dGPS measurements were taken at a later date, after sampling for this section was completed. We prioritized targeting peak CIE values from this section to ensure that the CIEs could be correlated together. This also helps avoid unnecessary uncertainty that could come from measuring dGPS elevations at incorrect locations in the section. Because the Basal/D1350 section does not have any CIEs at the base of the section, this would not affect our interpretation of the CIE levels.

Figure 4: for simplicity, delete -10 and -14 from all x-axis labels, it makes it easier to read

- We've made this change.

Figure 4: the panel b misses biostratigraphic information, while these would just relate to the isotope data as they do in panel a, right?

- We intentionally left this out of panel b. The biostratigraphic information (Biohorizon B, B-1, and B-2) have been directly tied in to the Bown et al. (1994) framework, however, they have not been directly tied to the new elevation framework or our carbon isotope results. We felt it would be misleading to show these biostratigraphic markers alongside the dGPS elevations because this would imply that we recreated the faunal analyses and found the same patterns with this new stratigraphic framework which we have not done. Instead, we wanted to show how our isotope data fit into the existing framework that includes the biostratigraphy (panel a) and then show the isotope results according to their elevation separately (panel b).

Line 330 is the PETM really an option? it seems a bit unnecessary to exclude the PETM as an option in the discussion I suppose? The PETM should be so much lower in stratigraphy? The early Eocene has a good bunch of CIEs to correlate to other than the PETM. Why are these Fifteenmile Creek CIEs not I1 and I2, their size would make that plausible at least.

- We mentioned the PETM because the values are more negative than what we see in the other CIEs in our sections and are more consistent with what has been seen in the

PETM. This is not, however, a plausible correlation, based on the biostratigraphy. The biostratigraphy also rules out correlation to I1 and I2, as mentioned in an earlier comment.

- Due to the ambiguous correlation between Basin Draw and the other sections, we no longer correlate the CIE in this record to any of the other CIEs in our record and its correlation remains unresolved.

Figure 5: magnetostratigraphy should be plotted along Fifteenmile Creek Composition Section

- We've added the magnetostratigraphy for Fifteenmile Creek to Figure 6 (Previously Fig. 5). Clyde et al. (2007) found that the original magnetostratigraphy from Fifteenmile Creek (Tauxe et al., 1994) contained some error in their placement of the C24r – C24n.3n reversal. The magnetostratigraphy from Clyde et al. (2007) was done ~20 km north in the Elk Creek Rim area of the Bighorn Basin (now shown in Fig. 1) and gives the best approximation of the position of the C24r – C24n.3n boundary in relation to these CIEs based on BCM levels in both places. It should be noted, however, that Clyde et al. (2007) also observed some relatively minor discrepancies with the BCM levels in the Elk Creek Rim compared to Fifteenmile Creek that were attributed to differences in sedimentation rates and cautioned against using these levels this far north. The correlation between our isotope results and the magnetostratigraphy should therefore be considered an approximation.

the distance between the centre of CIE2 and CIE3 in FCCSection, has decreased to just over 20m while nearly 40 m in North Fork section, the only section where the interval from CIE2 to CIE3 is measured in a continuous manner, how is the possible? It has a very big impact on the interpretation of the CIEs

- The spacing between CIE2 and CIE3 in the North Fork/Upper North Fork sections is ~20 m and this is the only place where these CIEs have been linked using a bed trace. We used this spacing to construct our composite section so this spacing is the same. There is ~40 m separating CIE1 and CIE3 in North Fork/Upper North Fork since we are interpreting the base of North Fork to be CIE1 (ETM2). We have added text to describe the additional uncertainty on the bed trace between the North Fork and Upper North Fork sections (lines 446-456). The differences in spacing between CIE2 and CIE3 in North Fork and Upper North Fork in these other frameworks (Fig. 4) compared to the Jacob's staff measurements (Fig. 3) suggest that there is some uncertainty of up to 20 meters in this part of our composite section.

Figure 5: it would be good to place the marine isotope stratigraphy to the far left as a baseline, including labels for hyperthermals, the MCP record next it, and the FCCS to the right.

- We've made this change. The figure with the composite section (Fig. 6) now shows the marine isotope stratigraphy and magnetostratigraphy along with the isotope stratigraphy and magnetostratigraphy for McCullough Peaks and Fifteenmile Creek.

Lines 360 to 366 is doubling of discussion with the 5.2 section.

- We've taken out the first part of this section that deals with spacing between CIEs and used this section to focus on magnitude differences.

Line 429-430: this sentence is written as if it is needed to have every CIE requires faunal change. I would be happy to believe that though I doubt whether we have much at hand at this stage to confirm anything close to that.

- We've changed the wording to "Such precise correlations could also be used to investigate other stratigraphic levels where we find CIEs, such as the 486 m BCM level, to see whether there is associated faunal change." (Lines 694-695)

---

## Author Response (AR2)

Our responses to the reviewer's comments are included below as bulleted points in black text. We have also made some other small changes to improve the manuscript, including adding high-resolution copies of the maps and field photos to the supplemental data repository and small spelling and editorial changes. These changes are all included in the marked-up version of the manuscript.

Comments from Reviewer #1:

1) In the introduction line 82: the authors could add a few papers that have looked at other hyperthermals continental sections including in the Tornillo Basin looking at ETM2 and H2 (Bataille et al. 2016; 2018), some work in the Pyrennees Eocene hyperthermal (Honneger et al. 2020), and the Provence Basin probably the ETM2 (Cojan et al. 2000)

- We have added these references as additional continental hyperthermal records to the introduction. We note that the Tornillo Basin records do not record CIEs associated with ETM2 and H2, but they do show sedimentological changes, including increased sand deposition, that are interpreted to be related to these hyperthermals.

2) In the method section: line 196 and appendix A. I am still a bit unclear as to when the "bed traces" vs. elevation vs. marker beds were used to tie up section/subsections. It would be nice to define "bed traces" because to me this is following a marker bed from one section to the next but I don't think this is the case for the authors as shown in Appendix A. In many of the figures in Appendix A (e.g. Fig. A1, A3...etc...), the authors show a black double-sided arrow indicating "bed traces" between beds that are clearly not similar stratigraphically? If the beds are not similar how do they correlate them? The authors usually show dGPS measurements at the base and top of each sub-section, but they do not say if those match? Is elevation then the main basis for correlation in those cases? In general, I find that the authors could be more explicit as to how/when they use elevation/marker bed/BCM correlations between sub-sections/sections. Also, I suggest the authors should use a different symbology in the figure of appendix A for "marker beds" tying (black double-sided arrows showing clear correlation between identifiable beds) vs. "bed traces + elevation (maybe black dashed double-sided arrows to underline uncertainty in tying).

- Any time we say "bed trace" we were following one or more marker beds to trace between outcrops. Typically, we traced the base of a marker bed (usually, prominent, red paleosols or the contact between two contrasting paleosols) to a new subsection. Sometimes a marker bed would become covered as we moved up section, and we would measure up to the next prominent marker bed and trace it into the next section. In most cases, though, we were able to measure back down to the original marker bed when more complete exposure was available. Occasionally, we would measure up to the base of a marker bed and trace this contact to a new subsection. In these cases, the marker bed was not originally shown in the subsection where we did not measure it in Figs. A1-A6 and the double-sided arrows were placed along the surface that was traced. This may have led to some confusion. We have added shaded boxes to show these bed traces more clearly in Figs. A1-A6 in addition to the double-sided arrows. We have also updated the text in the figure captions and in the Methods lines 193-198 to explain this.
- Differential GPS elevation was never used to correlate between subsections within a section. It was only used as a secondary check on the field correlations and to confirm the correlations between sections that were based on biostratigraphy and the existing stratigraphic framework from

Bown et al. (1994). The only time when dGPS elevations were used to adjust the stratigraphic levels is in panel B of Fig. 4, where the isotope values are shown according to their elevation. This is used to show that the elevations provide a reasonable correlation between sections that is similar to what we determined using the previous stratigraphic framework for the area and our own field measurements. We have added some text to the beginning of Appendix A (lines 546-550) to help clarify this.

3) Line 376 "terrane" should be terrain

- We made this change.

4) Dip measurements. Could the average and standard deviation of those be provided to get an idea of what close to zero means? I think that will be useful to have a better idea of the uncertainty of several km of distance when using dGPS.

- In most cases, we did not measure and record dip while measuring sections. Rather, when a marker bed could be visually traced to a point in the distance we would shoot a horizontal like with a clinometer to check to see if it was different from zero. It typically was not different than zero. If it was, we would adjust the clinometer on the Jacob Staff to account for dip in the direction we were measuring up section. Such a dip would be a directional dip, useful for measuring section. However, because of the shallowness of dip (or lack of dip) and the lack of 3D exposure, we did not attempt to determine attitudes (strike and dip). An average and standard deviation derived from such directional dips would not give an accurate representation of the dip in the region. Additionally, the error on these measurements would often be greater than the dip itself, so we do not report them.

5) Line 378: Also a problem if there is some structural displacement.

- We have added structural displacement to that line as another possible source of uncertainty in the dGPS measurements.

6) The idea of uniform zero dip across the area is not really validated by some of the conclusions of this paper particularly line 505-515. Could the authors provide an hypothesis for Bassin Draw not being aligned with the expected dGPS elevation? A fault? Greater distance with dip not equal to zero? What is the basis for suggesting a 20m downward move of the BD section, is it only correlation or is there other evidence (e.g., biostrat?)? Based solely on chemostrat, I could see a positive 25m upward move also fitting pretty well that is why I am asking.

- We have modified that section to include the possibility that the Basin Draw section could also be shifted upward. The biostratigraphy in the section is equivocal in terms of favoring one over the other. We also suggest the possibility of a fault running through Fifteenmile Creek as a potential mechanism for the misalignment. The possibility of a buried, east-west trending fault is supported by Kraus (1992).
- The text in lines 515-524 now reads, "It seems likely that the Basin Draw section localities should be shifted downwards by ~20 meters in the Bown et al. (1994) composite section. This would align the excursion at the top of Basin Draw with CIE1 and place the productive localities D-1454 and D-1460 within Biohorizon B. Alternatively, shifting the section up by ~25 meters would align the low isotope values near the base of the Basin Draw section with CIE1 and place the productive localities D-1454 and D-1460 closer to faunal event B–2. The biostratigraphy is equivocal in terms of favoring

one adjustment over the other, although the dGPS elevation supports the first option. Making one of these stratigraphic adjustments would likely strengthen the distinction between Biohorizon B and the subsequent faunal events B–1 and B–2 in a revised faunal analysis should further stratigraphic work demonstrate that this move is warranted. The possibility of a buried fault along Fifteenmile Creek could be responsible for some of the discrepancies in correlation across the creek. This would also be consistent with other observations of east-west trending fault activity during Eocene deposition (Kraus, 1992)."

---

## Author Response (AR3)

We have gone through the manuscript and edited a few minor spelling, grammatical, and formatting issues. We also made one small change to Figure 6 to make it clearer. The new version of this figure includes stratigraphic levels for two faunal events (B-1 and B-2) in the field area. Showing it this way helps to summarize one of our main conclusions – that the faunal events and CIEs occur near the same levels – and it does not change any of our original results, interpretations, or conclusions. The text now also includes a final DOI and link to the supplemental data housed in OSF.